# Transposon insertional mutagenesis of diverse yeast strains suggests coordinated gene essentiality polymorphisms

Piaopiao Chen [1], Agnès H. Michel[2] & Jianzhi Zhang [1✉]

Due to epistasis, the same mutation can have drastically different phenotypic consequences in different individuals. This phenomenon is pertinent to precision medicine as well as antimicrobial drug development, but its general characteristics are largely unknown. We approach this question by genome-wide assessment of gene essentiality polymorphism in 16 *Saccharomyces cerevisiae* strains using transposon insertional mutagenesis. Essentiality polymorphism is observed for 9.8% of genes, most of which have had repeated essentiality switches in evolution. Genes exhibiting essentiality polymorphism lean toward having intermediate numbers of genetic and protein interactions. Gene essentiality changes tend to occur concordantly among components of the same protein complex or metabolic pathway and among a group of over 100 mitochondrial proteins, revealing molecular machines or functional modules as units of gene essentiality variation. Most essential genes tolerate transposon insertions consistently among strains in one or more coding segments, delineating nonessential regions within essential genes.

[1] Department of Ecology and Evolutionary Biology, University of Michigan, Ann Arbor, MI 48109, USA. [2] Department of Biochemistry, University of Oxford, Oxford OX1 3QU, UK. ✉email: jianzhi@umich.edu

Even under the same environment, a mutation can cause different phenotypic effects in different individuals because of the genetic interaction (aka epistasis) between the mutation and the genetic background that varies among individuals[1–3]. The most extreme version of this phenomenon is when a gene is essential in one genetic background but nonessential in another such that a null mutation in the gene results in death and life in the two genetic backgrounds, respectively[4–6]. This phenomenon occurring in humans is a key reason why personalized or precision medicine is important and necessary in health care[7,8], is a cause of the genotype-dependent effects of cancer mutations[5,9–12], and may also underlie missing heritability[13]. The same phenomenon occurring in pathogenic microbes is pertinent to antimicrobial drug development because compared with genes whose essentiality varies among strains, those that are always essential have a greater potential as broad-spectrum antimicrobial drug targets[14–17].

Early interspecific comparisons estimated that over 20% of mouse orthologs of human essential genes are nonessential[18] and about 17% of fission yeast (Schizosaccharomyces pombe) orthologs of budding yeast (Saccharomyces cerevisiae) essential genes are nonessential[19]. Among bacterial taxa, orthologs of roughly 50% of Bacillus subtilis essential genes are not universally essential in Escherichia coli, Staphylococcus aureus, and Streptococcus sanguinis[20]. Along the same vein, intraspecific comparisons found that, between S288C and Sigma1278b, two S. cerevisiae strains with 0.32% genomic sequence divergence, 57 of more than 5000 genes examined are essential in only one of the strains[21]. Similarly, a genome-wide study reported extensive variation in gene essentiality among 18 representative E. coli strains[22]. Furthermore, many human genes have been found to be essential to some but not other cell lines[5,9–12]. In addition to surveys of wild types, recent years have seen systematic efforts in screening mutants where an otherwise essential gene becomes nonessential. This has been achieved for 17% of S. cerevisiae essential genes[23] and 27% of S. pombe essential genes[24].

Despite these findings of prevalent gene essentiality variation between and within species, a series of key questions are unanswered. For example, are different genes equally susceptible to essentiality changes in evolution? If not, what features determine the propensity for gene essentiality changes? Do different genes in the same genome independently change their essentiality in evolution? If not, what factors coordinate the essentiality changes of different genes? Answering these questions will help uncover general rules governing the genetic background dependency of mutational effects so will be instrumental to precision medicine and antimicrobial drug development. However, achieving this goal requires investigating gene essentiality variations at the genomic scale across a relatively large number of strains, which has become practical only recently thanks to the development of high-throughput methods such as pooled CRISPR interference (CRISPRi) screening and transposon insertional mutagenesis[22,25–30]. Transposon insertional mutagenesis has some advantages over CRISPRi and other CRISPR-based methods, primarily in requiring no knowledge of the genome sequence of the strain, being precise enough to inform intragenic regions that are essential, and most importantly avoiding the potential off-target effect of CRISPR genome editing[31]. In particular, Michel et al. developed saturated transposon analysis in yeast (SATAY) using a maize Activator/Dissociation (Ac/Ds) transposable element[25]. SATAY creates millions of cells each with an independent transposon insertion into the genome (except in the rare instance of transposition prior to induction). Because the number of transposed cells massively exceeds the number of genes in the genome, every gene is broken in multiple cells by independent transposition events. Subsequent sequencing of the flanking genomic regions of the inserted transposons (from viable cells) allows accurate determination of the insertion sites and essentiality of all genes in a strain in one single experiment.

In this work, we employ SATAY to assess gene essentiality variation across 16 S. cerevisiae strains (aka gene essentiality polymorphism) in order to address the series of questions aforementioned. We detect essentiality polymorphism in 9.8% of yeast genes, find that these genes and their protein products tend to have intermediate numbers of genetic and protein interactions, respectively, and discover a tendency for covariation of essentiality among genes encoding components of the same molecular machine or functional module.

## Results

**Genome-wide gene essentiality assessment by transposon insertional mutagenesis.** To apply SATAY[25] to a yeast strain, we first deleted its endogenous ADE2 and URA3 genes. We then transformed a plasmid containing the selectable marker URA3, inducible hyperactive transposase gene Ac under the control of the GAL1 promoter, transposon MiniDs, and ADE2 into the cells (Fig. 1a). MiniDs are located within and interrupt ADE2 on the plasmid. The cells were induced to express Ac on a synthetic defined (SD) medium with 2% galactose without adenine. Under the action of Ac transposase, MiniDs were excised out of the plasmid followed by the repair of the ADE2 gene; only those cells with repaired ADE2 could form colonies. The excised transposon was then integrated at a random position in the yeast nuclear genome (Fig. 1a).

We chose to investigate gene essentiality in 16 haploid, wild-type strains of S. cerevisiae that span diverse ecological and geographical origins (Fig. 1b, Supplementary Data 1). The between-strain genomic divergence measured by single nucleotide variants ranges from 0.06 to 0.6%, with a mean of 0.36%. For each strain, we collected approximately 1–3 million colonies to obtain a transposon insertion library nearly saturated at the gene level (Supplementary Data 2). All colonies were scraped off the plates and pooled, followed by overnight regrowth to saturation. The genomic DNA from all cells was extracted and digested with four-cutter restriction enzymes, followed by ligase-mediated circularization (Fig. 1a). Circular DNA was amplified by polymerase chain reaction (PCR) using transposon-specific outward-facing primers, and the PCR products were then sequenced on Illumina NextSeq 500 (see Methods). We aligned the obtained sequencing reads to the S288C reference genome, determined the precise transposon insertion sites, and counted the number of mapped reads per transposition for each library (Supplementary Data 2). On average, 0.34 million transposon insertions were identified per library, representing a density of one transposition for every 35 nucleotides if transpositions were evenly distributed along the genome. However, transpositions appeared to be enriched in centromeres (Supplementary Fig. 1a), probably owing to local transposition[32,33]. That is, the plasmid may cluster with yeast centromeres because it contains a yeast centromere segment[34]; consequently, the transposon excised from the plasmid tends to be inserted into centromeres due to spatial proximity. In addition, consistent with Michel et al.'s observation[25], we found through examining yeast nucleosome occupancy data[35] that transposons are preferentially inserted into nucleosome-free genomic regions (Supplementary Fig. 1b), perhaps because naked DNA is more accessible. Despite these insertional biases, with the exception of telomere repetitive regions, no large genomic regions (>10 kb) appeared transposon-free in our data (Supplementary Fig. 1c). Our observation is consistent with the reports that Ac/Ds transposons do not have a strong insertion site preference in maize or other organisms including yeast[25,33,36–38].

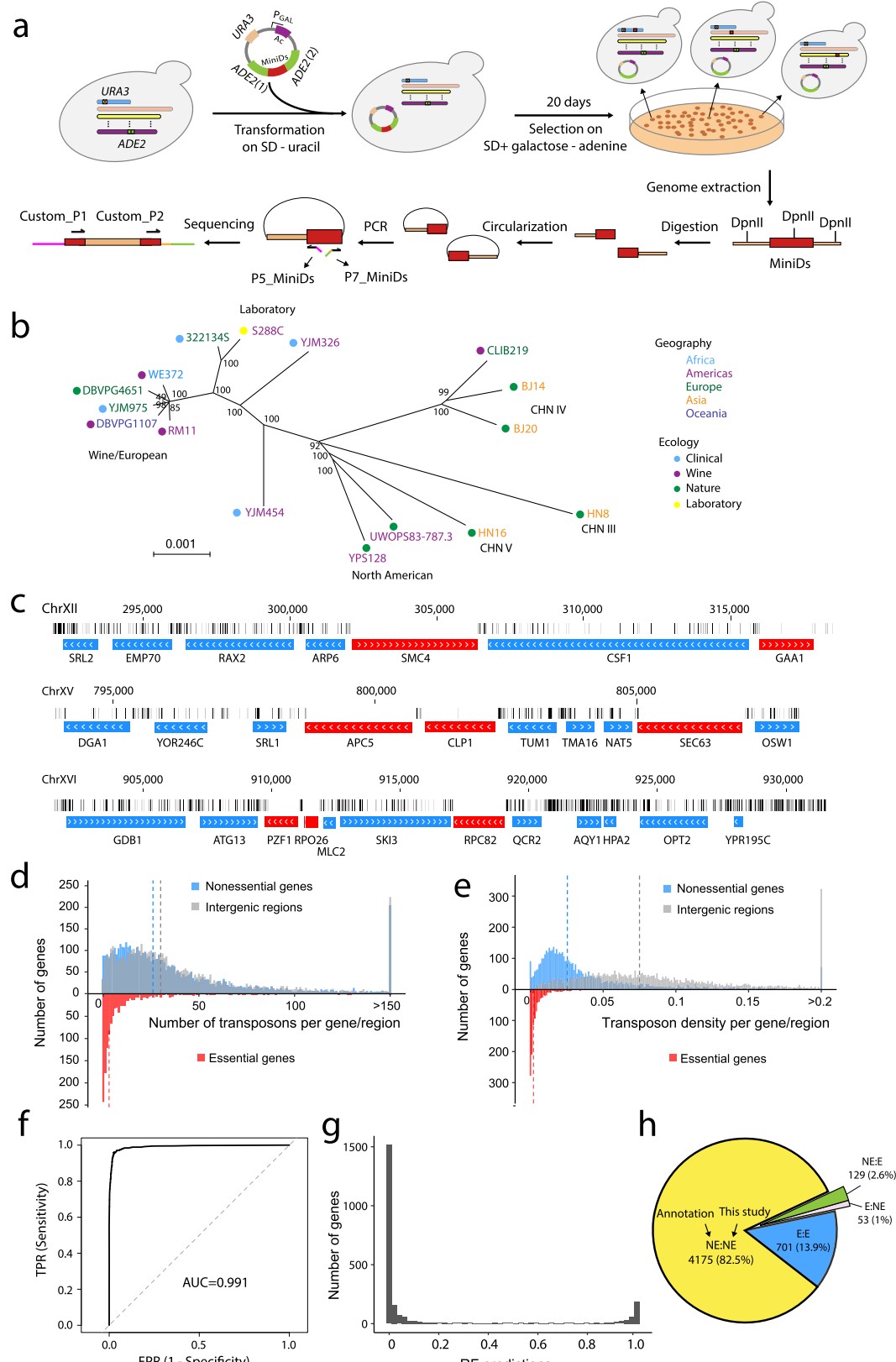

Our data from the strain S288C are generally consistent with annotations of essential and nonessential genes in the same strain that were based on individual heterozygous gene deletions followed by viability tests of haploid segregants[39]. The transposon map shows that the annotated nonessential genes are highly tolerant to transposons while essential genes are largely (but not completely) free of transposons in most of their coding regions (Fig. 1c). The median number of transposons (30) per intergenic region is similar to that (26) per annotated nonessential gene but much greater than that (3) per annotated essential gene (Fig. 1d). This pattern persists when transposon density, which is the number of transposon insertions per nucleotide of the (non-

**Fig. 1 Probing yeast gene essentiality polymorphism using transposon insertional mutagenesis. a** Outline of the experimental procedure. Endogenous *ADE2* and *URA3* genes are deleted (indicated by a cross) in all yeast strains. Plasmid pBK257 that carries the selectable marker *URA3*, inducible transposase gene *Ac* under the control of the *GAL1* promoter, and transposon *MiniDs* that interrupts *ADE2* is transformed into yeast cells. Cells grown on SD + galactose − adenine plates have the transposon excised from the plasmid and potentially randomly inserted into the yeast genome. Genomic DNA is extracted and digested with four-cutter restriction enzymes DpnII and NlaIII in parallel, followed by ligase-mediated circularization. Circular DNA is amplified using primer P5_MiniDs and P7_MiniDs to enrich transposon/chromosomal junction regions. Primer custom_P1 is for sequencing the flanking regions of *MiniDs*, while primer custom_P2 is for reading the 8-nucleotide index in the P7_MiniDs primer. **b** Neighbor-joining tree of the 16 *S. cerevisiae* strains used in the study. The tree is based on 179,416 single nucleotide polymorphisms (SNPs) identified in these strains and the scale bar represents 1 SNP per kb genomic sequence. Geographic locations and ecological origins of the strains as well as clade names (in black letters) are indicated. Bootstrap percentages are shown at interior branches. **c** Three examples of chromosomal segments showing genes and transposon insertions in S288C. The numbers after the chromosome number indicate genomic locations in nucleotides. Each vertical gray line represents one transposon insertion and the darkness of the line is proportional to the number of sequencing reads. Horizontal bars mark gene locations, with gene names provided below the bars and white arrows indicating transcriptional directions. Gene deletion-based essentiality annotations are shown by the color of the gene: red for essential and blue for nonessential. **d** Frequency distributions of the number of transposons per annotated essential gene (red), nonessential gene (blue), and intergenic region (gray) in S288C. Vertical dashed lines indicate medians of the corresponding distributions. **e** Frequency distributions of transposon density in annotated essential genes (red), nonessential genes (blue), and intergenic regions (gray) in S288C. **f** The receiver operating characteristic (ROC) curve analysis of predictions of S288C gene essentiality by the RF classifier. FPR false-positive rate, TPR true-positive rate, AUC area under the curve. **g** Frequency distribution of the RF predictions of essentiality for genes in the testing data. **h** Annotated and RF predicted essentialities of 5,058 genes in S288C. E essential, NE nonessential. Source data are provided as a Source Data file.

repetitive) genomic sequence, is compared among these regions (Fig. 1e). Throughout the analysis, unless otherwise noted, only coding regions were considered for genes.

**Gene essentiality classification using machine learning.** Because essential genes may harbor a small number of transposons, machine learning was previously used to distinguish essential from nonessential genes in transposon-based studies[38,40]. We here applied a similar approach to predict gene essentiality in S288C. Specifically, we constructed a random forest (RF) classifier with the following set of features of each gene using the transposon data: (1) number of transposons, (2) transposon density, (3) number of sequenced reads, (4) sequenced read density, (5) length of the longest transposon-tolerant coding region within the gene, (6) the above length divided by the gene length (aka freedom index), (7) number of transposons within the 100 nucleotides upstream of the gene, (8) transposon density in the gene divided by the transposon density in the surrounding 10,000 noncoding nucleotides (aka neighborhood index), and (9) number of gene segments without transposon insertion, each segment being one-tenth of the coding sequence of the gene (Supplementary Data 3). The classifier outputs a value between 0 and 1 that represents the probability that the gene of interest is essential. We treated gene deletion-based essentiality annotations in S288C as the ground truth. Note, however, that the gene deletion study used a rich medium whereas the present study used an SD medium. Hence, we removed from machine learning genes with potentially different essentiality in the two media, such as those found to be essential in a previous study under SD[25] but nonessential in the gene deletion study (see Methods). The resulting set of 4850 genes was used in training and testing the machine learning model (Supplementary Data 4).

The RF classifier was trained using 50% of the genes with a 10-fold cross-validation (i.e., training on 90% of the genes used and validating on the remaining 10%). The freedom index and neighborhood index made the most important contributions to the classification (Supplementary Fig. 2a). We used AUC (area under the receiver operating characteristic curve) to analyze the specificity and sensitivity of the classifier for the other 50% of the genes (testing data). The AUC reached 0.991 (Fig. 1f), meaning that the classifier scores a randomly chosen essential gene higher than a randomly chosen nonessential gene with a probability of 0.991. Furthermore, the RF outputs for the testing data are concentrated at 0 and 1 (Fig. 1g), indicating that the essentiality was predicted with

high confidence for the vast majority of genes. The features of each gene and the corresponding RF output are provided in Supplementary Data 4.

We divided the testing data into 20 bins, with their RF outputs in the ranges of 0–0.05, 0.05–0.10, …, and 0.95–1, respectively. As expected, the gene essentiality prediction is unreliable when the RF output is around 0.5 (Supplementary Fig. 2b). To identify the RF output corresponding to a 95% accuracy in essential gene prediction, we performed a polynomial regression across the 20 bins between the RF output and the fraction of essential genes ($f_{ES}$), finding that $f_{ES}$ reaches 0.95 when the RF output equals 0.903 (Supplementary Fig. 2c). Similarly, we found that $f_{ES} = 0.05$ when the RF output = 0.129 (Supplementary Fig. 2c). Focusing on genes with the RF output ≥ 0.903 yields a false discovery rate (FDR) of essential genes equal to 0.0096, and focusing on genes with the RF output ≤ 0.129 yields an FDR of nonessential genes equal to 0.0059. Together, 89.8% of the testing data belong to the above two groups of genes.

Using the above machine learning model and established thresholds, we predicted the essentiality of 5058 (87.1%) genes in S288C; SATAY-based gene essentiality was designated "undetermined" for the remaining 747 (12.9%) genes. Of the 5,058 genes with SATAY-based gene essentiality predictions, 4876 (96.4%) have the same predicted and annotated essentiality, 129 (2.55%) are predicted essential but annotated nonessential, and 53 (1.05%) are predicted nonessential but annotated essential (Fig. 1h). Several reasons can account for the prediction-annotation disparities. First, some genes nonessential in the rich medium used in the gene deletion study are expected to be essential in the medium used here, including auxotrophic genes (*PRO1*, *SER2*, and *SAC1*) and adenine biosynthetic genes (*ADE4*, *ADE6*, *ADE8*, and *ADE12*) (Supplementary Fig. 3a). Additionally, five annotated essential genes (*HIP1*, *MYO1*, *SPP381*, *NET1*, and *SEC3*) tolerate transposon insertions in the entire coding sequences (Supplementary Fig. 3b), which may also reflect condition/genotype-dependent gene essentiality. For example, *HIP1* encodes a histidine permease used for histidine uptake from the medium and is essential in *HIS3*-lacking strains such as the one used in the gene deletion study[25], but it is nonessential in the S288C strain that has *HIS3*. Second, 51 annotated essential genes overlap with other essential genes in coding sequences. However, in each of these focal genes, the non-overlapping part of the coding region tolerates transposon insertions (Supplementary Fig. 3c, Supplementary Data 5). Indeed, 13 of them are predicted nonessential,

two are predicted essential, and the rest are undetermined. Last, some annotated essential genes contain such small transposon-free regions that our model considers them transposon-free by chance so calls them nonessential (Supplementary Fig. 3d). Note that the above prediction-annotation disparaties should have minimal impacts if any on our subsequent comparison among yeast strains that are based entirely on the SATAY data.

**Gene essentiality polymorphism in yeast**. If gene essentiality is conserved across *S. cerevisiae* strains, we should observe a depletion of transposon insertions across strains for genes annotated as essential in S288C. Indeed, in each strain, we observed a significant reduction in the number of insertions in genes annotated essential in S288C relative to that in genes annotated nonessential in S288C and that in intergenic regions (Fig. 2a, Supplementary Fig. 4). The complete map of our transposon insertions in the 16 strains is available at http://genome-euro.ucsc.edu/s/Piaopiao/samples_16strains. Because of the lack of gene deletion-based essentiality annotations for the 15 non-S288C strains, we predicted gene essentiality in each of these strains using the machine learning model and thresholds established in S288C adjusted for the difference in the number of transposons between the focal strain and S288C (see Methods). The respective features of each annotated gene used in machine learning predictions are listed in Supplementary Data 6 for each strain.

The 16 strains analyzed show a total of three incidents of aneuploidy, including an extra Chromosome I in the WE372 strain, an extra Chromosome III in RM11, and an extra left arm of Chromosome VII in YJM326, evident from the approximately doubled transposon densities in the involved chromosomes (Supplementary Fig. 5a). The creation of an extra chromosome (prior to transpositions) renders all essential genes on the chromosome tolerant to transposon insertions (Supplementary Fig. 5b). While these incidents represent genuine gene essentiality changes, their pattern (loss of essentiality along a chromosome) and mechanism (gain of a chromosome) are clear. Thus, the essentiality of the genes in the corresponding chromosomes of aneuploid strains is marked "undetermined" so that our subsequent analysis could concentrate on other patterns and mechanisms of essentiality changes. In addition to RM11, strain DBVPG4651 showed a virtually universal tolerance to transposon insertions in the left arm and part of the right arm of Chromosome III (Supplementary Fig. 5c). Although we observed no obvious doubling of transposon densities in Chromosome III of DBVPG4651 (Supplementary Fig. 5d), it is extremely unlikely that all of these essential genes have simultaneously switched to nonessential genes via a non-aneuploidy mechanism. Thus, to make our analysis conservative, we also marked the essentiality of the genes on Chromosome III of DBVPG4651 "undetermined".

The essentiality of 5814 genes was determined in at least one of the 16 strains by SATAY, among which 5813 genes had essentiality determined in at least two strains (Supplementary Data 6). Of these genes, 567 have switched their essentiality in at least one strain and are referred to as polymorphic essential genes (Supplementary Data 7; see examples in Fig. 2b); the rest are referred to as monomorphic essential or monomorphic nonessential genes. For these 567 genes, we then relaxed the thresholds to $f_{ES} = 0.75$ and 0.25 to allow essentiality predictions in more strains. Even with these relaxed thresholds, predictions are still expected to be fairly reliable. For example, in S288C, they correspond to an FDR of 0.046 for essential genes and an FDR of 0.009 for nonessential genes. Among these 567 genes, 85 overlap with the genes differing in essentiality between budding yeast and

fission yeast, significantly more than the random expectation of 39 overlaps ($P < 0.0001$, permutation test). This observation suggests that the propensity for essentiality changes varies among genes and that the gene-specific propensity is relatively stable evolutionarily.

Examining the among-strain variation in gene essentiality on the phylogeny of the 16 yeast strains, we noticed that, for many genes, essentiality must have switched multiple times along the tree because the strains with the same essentiality state do not form a monophyletic group (Fig. 2c). To quantitatively analyze the rate of essentiality changes of individual genes, we used the parsimony principle to count the number of essentiality changes per gene along the 16-strain phylogeny. If this rate is equal among genes, the number of essentiality changes per gene should follow a Poisson distribution, whose variance equals the mean. The actual distribution observed from all genes is overdispersed ($P = 1.9 \times 10^{-232}$, a chi-squared test of the null hypothesis that the observed variance equals the mean), with overrepresentations of genes in the categories of no changes and >2 changes (Fig. 2d). Overdispersion was also evident ($P < 1.9 \times 10^{-232}$) under the assumption of a star phylogeny of the 16 strains. Introgression can make the assumed strain phylogeny incorrect for some genomic regions, which could inflate the inferred number of essentiality changes along the phylogeny. To exclude the possibility that the observed overdispersion is entirely due to introgression, which is common in *S. cerevisiae* natural populations[41], we used single nucleotide polymorphisms (SNPs) observed in the same 16 strains as a control because SNPs are subject to the same introgression. Specifically, we randomly sampled the same number of SNPs as the number of genes exhibiting essentiality polymorphism, requiring that the allele frequencies of the sampled SNPs match the essentiality frequencies of these 567 genes in the 16 strains. This was repeated to obtain 1000 random sets of control SNPs. We found the mean number of essentiality changes per gene to be significantly greater than the mean number of nucleotide changes per SNP in every set of control SNPs (Fig. 2e). Hence, introgression cannot explain the overdispersed distribution of the number of gene essentiality changes, demonstrating a genuine variation of the rate of essentiality changes among genes. Substitution rate variation among amino acid sites in a protein is usually modeled by a gamma distribution; the smaller the gamma shape parameter $\alpha$, the greater the rate variation[42]. Using the same concept and method[43], we estimated that $\alpha = 0.113$ for gene essentiality changes, smaller than the corresponding values for amino acid substitutions in all 51 vertebrate nuclear genes previously examined[43], indicating a very strong variation of the rate of essentiality changes among genes.

**Protein/genetic interactions and polymorphic gene essentiality**. Compared with proteins participating in few protein-protein interactions, those with many interactions are thought to be more important in function or have a higher chance to engage in at least one essential interaction so are more likely to be essential[44,45]. Indeed, the number of protein interaction partners ($N_{PI}$) was found significantly greater for essential genes than nonessential genes when the gene essentiality and $N_{PI}$ data from S288C were examined two decades ago[44]. Yeast protein interactions are generally evolutionarily conserved[46], but gene essentiality is not generally conserved, as shown here and before[19,21]. Interestingly, we found that genes with polymorphic essentiality are between monomorphic essential genes and monomorphic nonessential genes in terms of $N_{PI}$ (Fig. 3a). This observation is easy to understand if having a greater $N_{PI}$ increases a gene's probability to be essnetial[44,45]. That is, because of the general

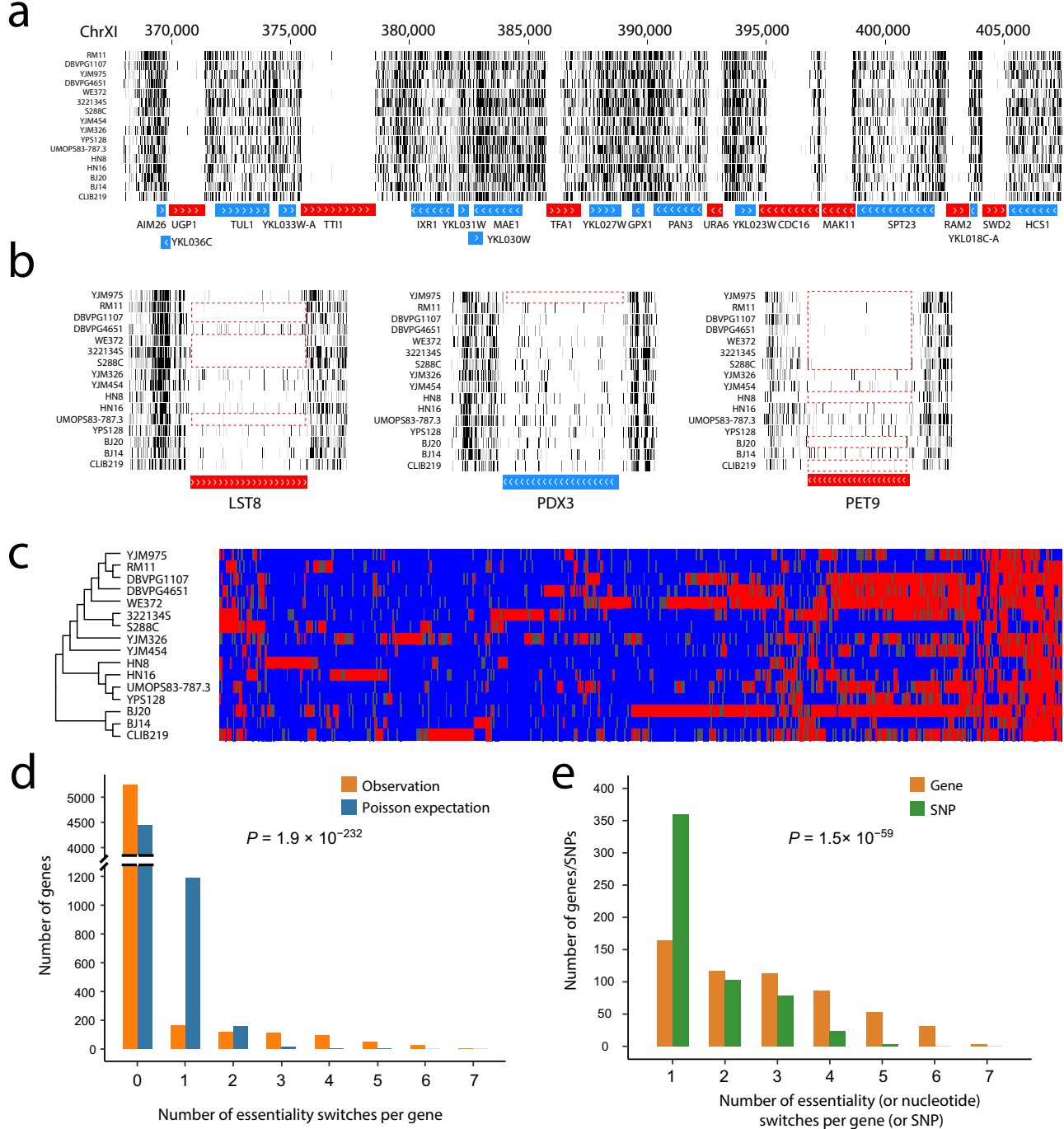

**Fig. 2 Gene essentiality polymorphism in yeast. a** A typical example of a genomic region with transposon insertions in each strain. Stain names are shown on the left side of the panel. Eight annotated essential genes (red bars) have part or all of their coding regions virtually transposon-free, while the 15 annotated nonessential genes (blue bars) tolerate transposons in all of their coding regions. Horizontal bars mark gene locations, with gene names provided below the bars and white arrows indicating transcriptional directions. Each vertical gray line represents one transposon insertion and the darkness of the line is proportional to the number of sequencing reads. **b** Three examples of gene essentiality polymorphism in the 16 yeast strains. The red boxes highlight strains where the gene is (virtually) transposon-free. **c** Gene essentiality polymorphism in the 16 strains. Each column represents a gene while each row represents a strain. Red, essential genes; blue, nonessential genes; gray, essentiality undetermined. The phylogeny of the 16 strains as in Fig. 1b is shown on the left, but the branches are not drawn to scale. **d** Distribution of the observed number of essentiality changes per gene along the phylogeny of the 16 strains, compared with the Poisson distribution with the same mean. *P*-value is from a one-tailed chi-squared test comparing the variance of the observed distribution with the corresponding Poisson variance. **e** Distribution of the observed number of essentiality changes per gene along the 16-strain phylogeny for the 567 genes exhibiting essentiality polymorphism, compared with the corresponding distribution of the number of nucleotide switches per SNP along the phylogeny for 567 randomly picked SNPs with levels of polymorphism matching levels of gene essentiality polymorphism. *P*-value from the two-tailed Wilcoxon signed-rank test that compares the mean ranks of the two distributions is presented. We perform 1000 replications of sampling of 567 SNPs and find $P < 2.9 \times 10^{-52}$ in every replication. Source data are provided as a Source Data file.

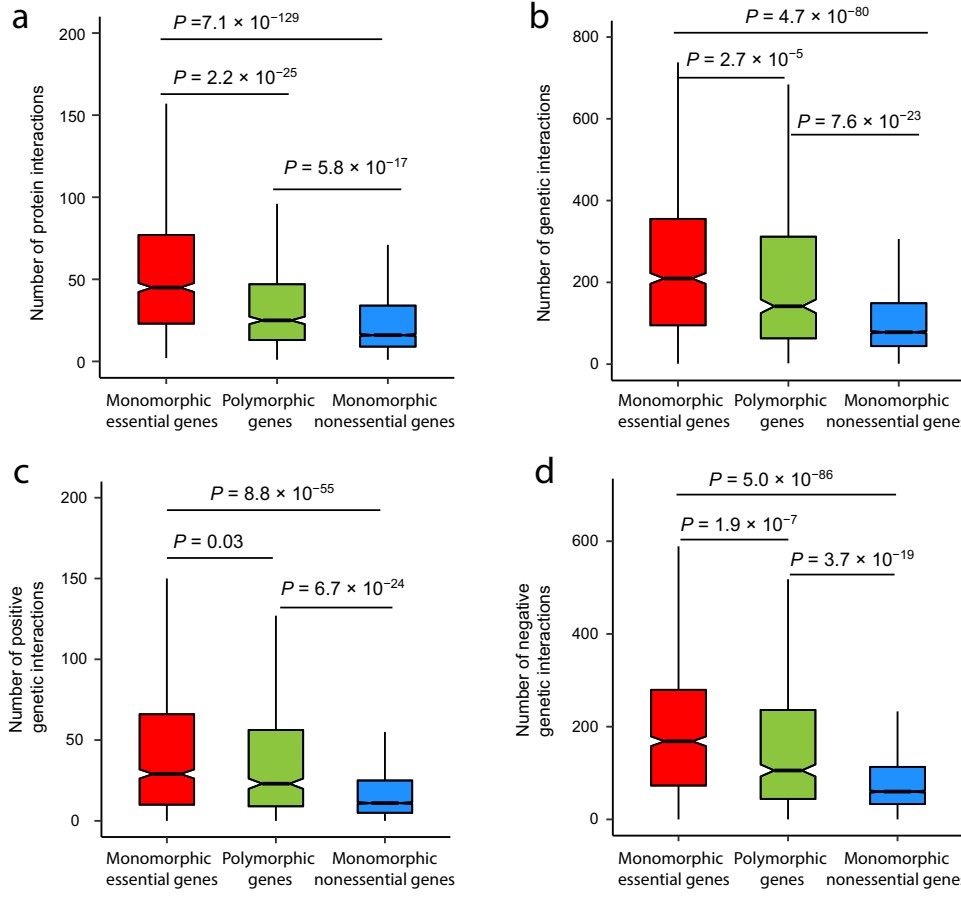

**Fig. 3 Numbers of protein and genetic interactions per gene. a** Number of protein-protein interaction partners per gene. **b** Number of genetic interaction partners per gene. **c** Number of positive genetic interaction partners per gene. **d** Number of negative genetic interaction partners per gene. In each box plot, the lower and upper edges of a box represent the first ($qu_1$) and third ($qu_3$) quartiles, respectively, the horizontal line inside the box indicates the median ($md$), and the whiskers extend to the most extreme values inside inner fences, $md \pm 1.5(qu_3 - qu_1)$. Notches show the 95% confidence interval of the median. *P*-values are from two-tailed Wilcoxon rank-sum tests. Polymorphic genes ($n = 567$) refer to those that are essential in some but not other yeast strains examined. Monomorphic essential genes ($n = 1046$) are essential in all examined strains, whereas monomorphic nonessential genes ($n = 4201$) are nonessential in all examined strains. Source data are provided as a Source Data file.

conservation of protein interactions[46], the $N_{PI}$ of a gene has a small variation among individuals within a species. Hence, only when $N_{PI}$ is at an intermediate level can a small variation in $N_{PI}$ move it between essential and nonessential zones and create gene essentiality polymorphism.

Because the number of genetic interactions ($N_{GI}$) was also reported to be significantly higher for essential genes than nonessential genes[47,48], we examined the $N_{GI}$ for polymorphic essential genes. Again, polymorphic essential genes are between monomorphic essential and monomorphic nonessential genes in $N_{GI}$ (Fig. 3b). This trend holds for both positive (Fig. 3c) and negative (Fig. 3d) genetic interactions despite that negative interactions indicate gene functional relationships better than positive interactions[47,49]. Because measuring genetic interaction typically involves deleting genes, the $N_{GI}$'s of essential and nonessential genes were measured in S288C by different strategies[47]. Hence, it is possible that their $N_{GI}$'s are not directly comparable. We thus compared monomorphic with polymorphic genes among S288C essential genes and confirmed that $N_{GI}$ is higher for the former than the latter (Supplementary Fig. 6). Similarly, we verified that $N_{GI}$ is lower for monomorphic than polymorphic genes among S288C nonessential genes (Supplementary Fig. 6). Note that the $N_{PI}$'s of essential and nonessential genes can be fairly compared because measuring protein interactions does not involve deleting genes.

Together, the above results show that polymorphic essential genes tend to have intermediate numbers of protein and genetic interactions when compared with monomorphic essential genes and monomorphic nonessential genes. Hence, a focus can be placed on genes with intermediate numbers of protein and genetic interactions if one is interested in identifying genes with polymorphic essentiality.

**Correlated essentiality changes among genes.** We performed a Gene Ontology (GO) analysis of the biological processes of the 567 genes that exhibit essentiality polymorphism (against the background of the 5,814 genes examined) and found significant enrichment with mitochondrial gene expression, mitochondrial translation, mitochondrial genome maintenance, and several metabolic processes (Supplementary Fig. 7, Supplementary Data 8). The functional enrichment prompted us to investigate whether essentiality changes are coordinated between different genes in the same genome. That is, we correlated the gene essentiality status for each pair of the 567 genes across the 16 strains after considering their phylogenetic relationships[50] (Supplementary Data 9). We then used the correlation coefficients to perform a gene clustering analysis that, under a height cutoff, identified groups of genes with high within-group inter-correlations (Supplementary Fig. 8). Below we describe the findings under the cutoff of height = 7 to strike a balance between

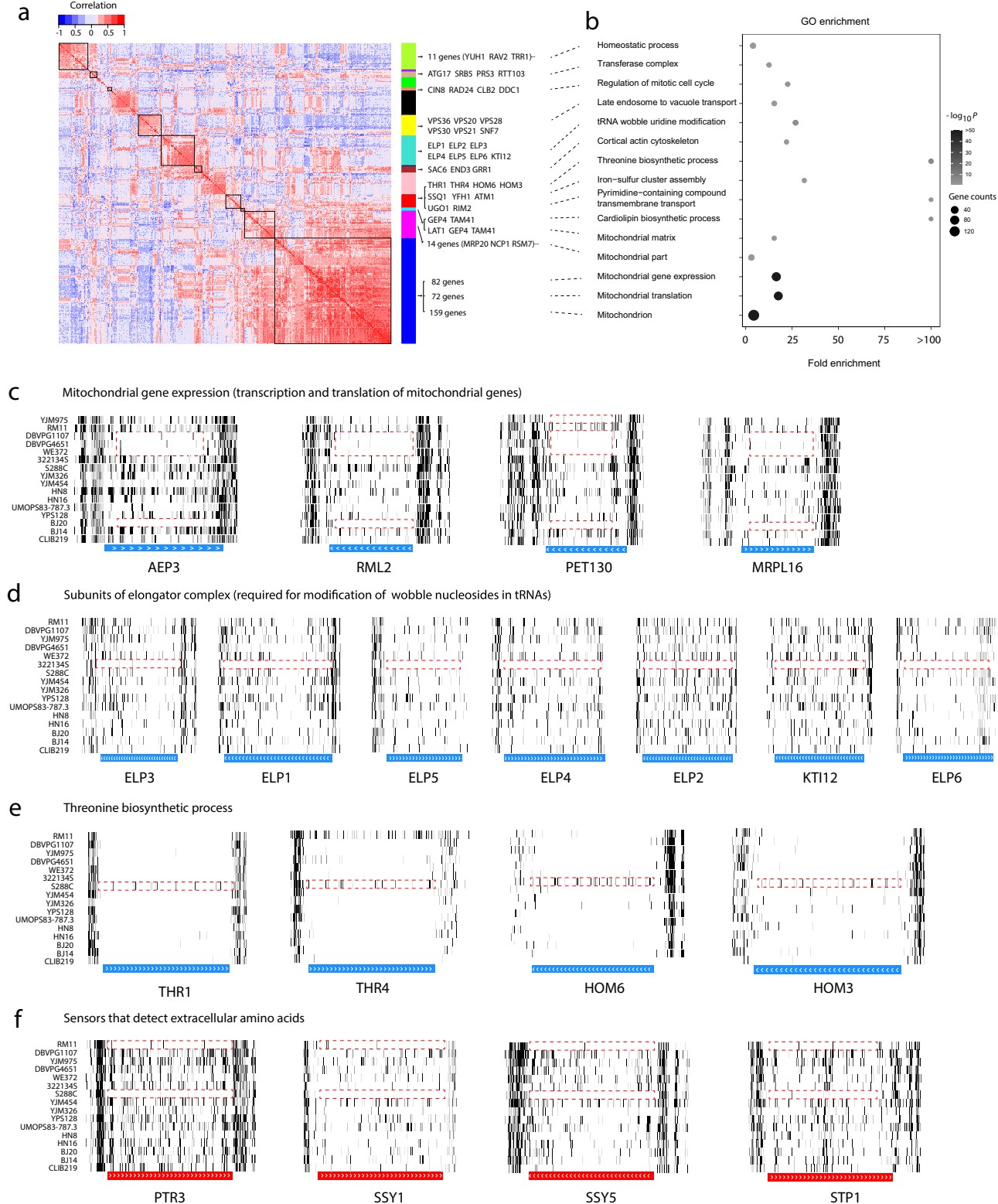

the number of gene groups and specificity of each group, but the biological findings are overall consistent under height = 6–8 (Supplementary Fig. 8). It is clear that each group comprises genes with strong among-strain covariation in essentiality (Fig. 4a), and these genes often show significant GO term enrichment (Fig. 4b). For example, in the largest group identified (lowest box in Fig. 4a), which includes 198 genes, 159 have their protein products located in the mitochondrion. Genes in this

group are mostly essential in strains DBVPG1107, DBVPG4651, WE372, and BJ20 (the first three being European/wine strains while the last being Chinese strain), but mostly nonessential in the other strains (Fig. 4c). Interestingly, Kim et al. reported 95 genes related to the mitochondrial function that have opposite gene essentiality between fission yeast and budding yeast[19], 53 of which belong to this group. Clearly, mitochondrial functions are frequently involved in yeast intraspecific and interspecific gene

**Fig. 4 Coordinated gene essentiality polymorphisms. a** Correlation in gene essentiality changes along the yeast phylogeny between any two of the 567 genes with essentiality polymorphism, followed by clustering analysis. In the heat map, each column/row represents a gene. Boxes along the diagonal and the corresponding vertical bars next to the heat map show gene groups identified from the clustering analysis to have high overall within-group correlations. Names are given for genes in the group that belong to the enriched GO term in panel **b** (connected by a dotted line) unless there are too many genes, in which case we provide the corresponding gene number that may be followed by names of a few representative genes. **b** Enrichment of GO terms for genes in each gene group in panel a. Gene count, number of genes in the gene group that belong to the enriched GO term. The shade of a circle indicates the statistical significance of GO enrichment measured by the *P*-value from the hypergeometric test adjusted for multiple testing using the Bonferroni correction. **c**–**f** Transposon insertion maps in the 16 strains for four genes functioning in mitochondrial gene expressions (**c**), the seven genes encoding all members of the elongator complex and a cofactor (**d**), four genes in the threonine biosynthetic pathway (**e**), and four genes encoding components of the SPS plasma membrane amino acid sensor system (**f**). Note that *THR4* is located on Chromosome III, so its tolerance to transposons in RM11 and DBVPG4651 is also aneuploidy-related. Horizontal bars mark gene locations, with gene names provided below the bars and white arrows indicating transcriptional directions. Gene deletion-based essentiality annotations in S288C are shown by the color of the gene: red for essential and blue for nonessential. Source data are provided as a Source Data file.

essentiality changes. Notably, the loss of mitochondrial DNA is non-lethal to many *S. cerevisiae* strains, a phenomenon known as petite-positive, but the loss becomes lethal (known as petite-negative) upon the inactivation of genes encoding F1-ATPase subunits, ATP/ADP carrier, i-AAA protease complex, phosphatidylglycerophosphate synthase, or mitochondrial protein import components[51–53]. It is possible that DBVPG1107, DBVPG4651, WE372, and BJ20 are petite-negative just like fission yeast.

The elongator complex including six subunits (ELP1-6) was originally described as a transcription elongation factor, but increasing evidence suggests that its primary function is to modify tRNAs at their wobble base position[54,55]. This complex is highly conserved from yeast to human, and deficiencies in the human elongator give rise to severe pathological defects such as familial dysautonomia, intellectual disabilities, and other neurological disorders[56]. While all six complex subunits as well as the cofactor KTI12 are nonessential in the vast majority of the yeast strains surveyed, they are all essential in the clinical strain 322134S isolated from human throat/sputum (Fig. 4d). Interestingly, ELP genes are essential to laboratory strains W303 and SEY6210, which are highly closely related to S288C, at 37 °C but not 30 °C[57–59], suggesting the possibility of alternative elongator that works at 30 °C but not 37 °C. Genes encoding the alternative elongator may be inactivated or lost in strain 322134S that lives at 37 °C; consequently, ELP genes become essential to the strain if the temperature lowers to 30 °C. Four genes (*THR1*, *THR4*, *HOM2*, and *HOM6*) involved in threonine biosynthesis are essential for growth in most strains but are nonessential in S288C (Fig. 4e). The SPS (Ssy1-Ptr3-Ssy5) amino acid sensing pathway and its transcriptional regulator Stp1 are nonessential in most strains but essential in RM11 and S288C (Fig. 4f). This polymorphism arises because RM11 and S288C used in our study are leucine auxotrophic but defective SPS signaling impairs leucine uptake[60].

The above examples suggest the possibility that gene essentiality generally changes concordantly among members of the same protein complex or metabolic pathway during evolution. Below, we first test at the genomic scale the above hypothesis regarding protein complexes. We noticed that 13.1% of genes encoding protein complex components exhibit essentiality polymorphism, significantly greater than that (8.3%) of other genes (Fig. 5a). We divided all pairs of the 567 genes exhibiting essentiality polymorphism into four groups. Group I comprises pairs of genes that encode two components of the same protein complex. Group II comprises pairs of genes that encode components of different protein complexes. In Group III, each gene pair contains one gene that encodes a protein complex member and the other that does not encode any protein complex member. Group IV consists of gene pairs that do not encode protein complex members. We found the between-gene correlations in essentiality changes presented in

Fig. 4a to be significantly greater for Group I than for each of the other groups (Fig. 5b), strongly supporting the hypothesis that essentiality tends to change concordantly among members of the same protein complex. Supplementary Data 10 lists the 47 complexes each having at least two members exhibiting gene essentiality polymorphism.

Among genes encoding metabolic pathway components (see Methods), 14.2% exhibit essentiality polymorphism, significantly greater than the corresponding value (9.1%) among other genes (Fig. 5c). We similarly divided the 567 genes with essentiality polymorphism into four groups based on their involvement in metabolic pathways. Again, the between-gene correlations in essentiality changes presented in Fig. 4a are significantly greater for Group I than for each of the other three groups (Fig. 5d), strongly supporting our hypothesis that essentiality tends to change concordantly among components of the same metabolic pathway. Supplementary Data 11 lists the 37 metabolic pathways each having at least two components exhibiting gene essentiality polymorphism.

The over-representation of metabolic pathway components and protein complex components in genes exhibiting essentiality polymorphism may be due to a large number of indispensable redundancies in the metabolic network and protein complexes[61,62]. Mutations that remove these redundancies may have fitness effects only in some environments so can spread in other environments. Consequently, genes involved in these pathways/complexes are prone to essentiality polymorphism.

**Most essential genes tolerate transposon insertions in at least one coding segment**. One advantage of using transposon insertion mutagenesis over gene deletion or CRISPRi in assessing gene essentiality is that it provides information about essential and nonessential genomic regions that is independent of existing annotations of genic regions. Indeed, several studies reported that some annotated essential genes tolerate transposon insertions in one or more segments of their coding regions[25,63], allowing delineating nonessential parts of essential genes. However, the prevalence and conservation of this phenomenon across *S. cerevisiae* strains are unclear. Our data provide an opportunity to estimate the prevalence of this phenomenon as well as the consistency of the boundaries between essential and nonessential genic parts among strains. We divided the coding sequence of a gene into ten equal-length segments and examined the distribution of transposon insertions among the ten segments. In S288C, 578 annotated essential genes each have at least two transposon insertions in at least one of the segments. Among these genes, we observed an average of 418 genes (72%) that show the same phenomenon in each of the other 15 strains, suggesting that this phenomenon is shared among yeast strains.

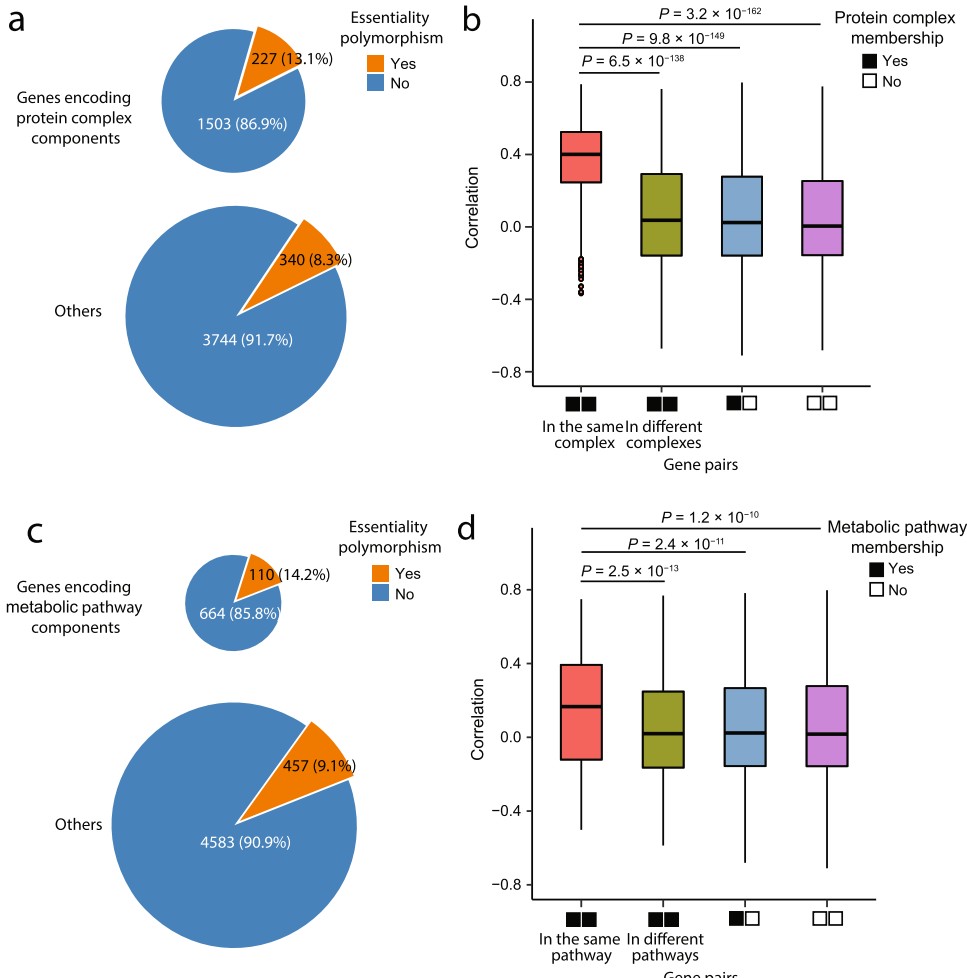

**Fig. 5 Coordinated essentiality changes of genes encoding members of the same protein complex or metabolic pathway. a** Genes encoding protein complex components are more likely than other genes to exhibit essentiality polymorphism ($P = 1.74 \times 10^{-8}$, one-tailed chi-squared test). Pie size is proportional to gene number. **b** Correlations presented in Fig. 4a are greater for pairs of genes encoding components of the same protein complex ($n = 643$ gene pairs) than those encoding components of different complexes ($n = 24,782$), those including only one gene encoding a complex component (indicated by one black square and one white square on the x-axis; $n = 77,066$), and those not encoding complex components (indicated by two white squares on the x-axis; $n = 57,970$). **c** Genes encoding metabolic pathway components are more likely than other genes to exhibit essentiality polymorphism ($P = 7.07 \times 10^{-6}$, one-tailed chi-squared test). Pie size is proportional to gene number. **d** Correlations presented in Fig. 4a are greater for pairs of genes encoding components of the same metabolic pathway ($n = 434$) than those encoding components of different pathways ($n = 5561$), those including only one gene encoding a pathway component ($n = 50,270$), and those not encoding pathway components ($n = 104,196$). In the box plots of panels b and d, the lower and upper edges of a box represent the first ($qu_1$) and third ($qu_3$) quartiles, respectively, the horizontal line inside the box indicates the median ($md$), the whiskers extend to the most extreme values inside inner fences, $md \pm 1.5(qu_3 - qu_1)$, and the dots represent values outside the inner fences (outliers). P-values are from two-tailed Wilcoxon rank-sum tests. Source data are provided as a Source Data file.

To determine more accurately the number of essential genes tolerating transposon insertions using the information from all 16 strains, we required a segment to tolerate at least one transposon insertion in at least 9 strains for it to be designated transposon-tolerating. We found that among the 1074 annotated essential genes with monomorphic essentiality, 669 genes (62.3%) have at least one transposon-tolerating segment, and 185 genes (17.2%) have at least three transposon-tolerating segments (Fig. 6a). Detailed information about transposon insertions in each segment is provided in Supplementary Data 12. The number of transposons in essential genes declines from outlying to central segments, especially from the 3′ end (Fig. 6b). This is in sharp contrast to nonessential genes where transposons are more or less uniformly distributed along the ten segments (Fig. 6c).

Figure 6d shows examples of essential genes where only the N-terminus, C-terminus, a continuous internal region, or several discontinuous regions are transposon-free. Because the *MiniDs*

transposon contains multiple stop codons in each reading frame, a gene with a *MiniDs* insertion in the coding region must be truncated. That cells with such truncated essential genes are nonetheless viable could be due to one or more of the following reasons. First and foremost, the protein can remain functional if the truncation occurs at the 3′ end of the coding region because most of the codons of the gene have been successfully translated. This explanation is consistent with the observation that transposons in essential genes are enriched in the 3' most 10-20% of the coding sequence (Fig. 6b). The successful removal from *TAF3* and *PRP45* of the 3′ regions that tolerate transposon insertions[25] supports this explanation. Second, that translation often starts from one of multiple alternative start codons[64] renders an essential gene tolerant to transposon insertions in the 5′ end of its coding region. Third, if a gene tolerates transposons in both its 3′ and 5′ coding regions respectively for the above two reasons, it may contain only an internal transposon-free region. Fourth, some genes have

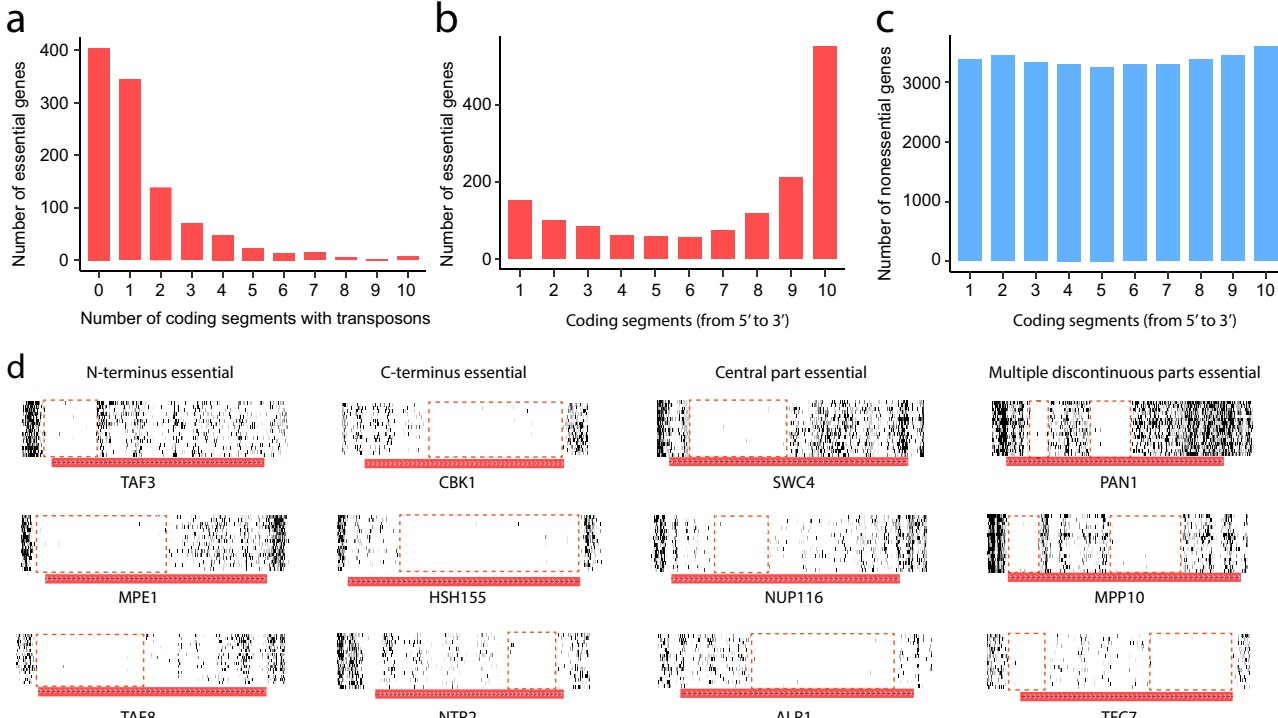

**Fig. 6 Most essential genes contain transposon-tolerant coding segments. a** Most essential genes tolerate transposon insertions in at least one coding segment. Each segment constitutes one-tenth of the coding region, and a segment is considered tolerant to transposon insertions only when this is true in at least nine strains. **b** Number of monomorphic essential genes that tolerate transposons in each coding segment. **c** Number of monomorphic nonessential genes that tolerate transposons in each coding segment. **d** Examples of genes that are essential only in the N-terminus (first column), only in the C-terminus (second column), only in the central part (third column), and in multiple discontinuous parts (fourth column). Horizontal bars mark gene locations, with gene names provided below the bars and white arrows indicating transcriptional directions. Source data are provided as a Source Data file.

multiple discontinuous transposon-free regions. We speculate that this phenomenon may indicate that the gene encodes two or more proteins that span different regions of the annotated coding region of the gene, resembling prokaryotic operons. Indeed, a growing number of polycistronic mRNAs (multiple independent proteins encoded on a single molecule of mRNA) have been identified in fungi[65], plants[66,67], green algae[68], nematodes[69], and flies[70] over the past decades. Last, it is also possible that multiple essential non-coding RNAs are encoded by these multiple discontinuous transposon-free regions of an annotated gene. Future studies are needed to test these hypotheses.

## Discussion

Using SATAY, we created and probed millions of transposon insertions in the genomes of 16 *S. cerevisiae* strains, providing high-resolution genome-wide data for investigating gene essentiality polymorphism. We found 567 genes exhibiting essentiality polymorphism. The number of genes with variable essentiality between two strains varies from 62 to 258, with a mean of 136. Thus, the previously reported number of 57 gene essentiality switches between S288C and Sigma1278b is atypically low, which could be due to the use of a different experimental method in the previous study[21]. Very recently, Parts et al. crossed a collection of temperature-sensitive (TS) mutant alleles of 580 essential genes in strain S288C with 10 diverse yeast strains and isolated about 60,000 segregants carrying the TS allele and a mosaic of S288C and other parents[71]. By measuring the fitness at the restrictive temperature, the authors found that 26% (149/580) of these essential genes in S288C could become nonessential in the genetic background of at least one of the 10 strains. This fraction is much higher than our estimate that 3.5% of essential genes in S288C

can become nonessential in at least one of the 16 strains surveyed here. This disparity may be in part because the essential genes were not deleted or truncated in Parts et al.'s study and the leaky expression from a TS promoter may be sufficient to support cell growth for some essential genes in some genetic backgrounds. Additionally, the TS alleles may be temperature-sensitive in some but not all genetic backgrounds.

By systematic analysis of the genetic context-dependency of the essentiality of 728 essential genes in S288C, van Leeuwen et al. reported 124 genes that could become nonessential upon mutations in the strain[23]. We observed that 38 essential genes in S288C can become nonessential in at least one of the 15 non-S288C strains surveyed here. Among the 38 genes, 18 overlap with the 124 genes reported[23], significantly more than the random expectation of 4 overlaps ($P < 0.0001$, permutation test). In 12 of these 18 cases, the modifier genes were identified[23], including six cases where the modifiers carried loss-of-function mutations and four cases where the modifiers carried gain-of-function mutations. However, by examining the SNPs and indels of the 16 strains in our study, we did not observe any loss-of-function mutations in those modifiers of the former 6 genes, nor did we observe the same gain-of-function mutations in those modifiers of the latter 4 genes. Hence, the genetic mechanisms of essentiality polymorphism in nature likely differ from those identified through genetic screening in the lab[23].

We found the rate of essentiality changes to vary substantially among genes. Most genes are monomorphic in essentiality, but a small fraction of genes have experienced multiple (up to seven) essentiality switches along the tree of the 16 strains. Interestingly, polymorphic essential genes sit between monomorphic essential and monomorphic nonessential genes in terms of their numbers of protein and genetic interactions, as if they are partially

essential. We found essentiality changes to be highly concordant for many pairs or groups of genes with functional relationships, the most prominent being a group of over 100 genes with mitochondrial functions. Furthermore, genes encoding members of the same protein complex or components of the same metabolic pathway tend to change essentiality concordantly. These observations suggest that molecular machines or functional modules are units of essentiality changes and polymorphisms. This insight, further confirmed in other species, could be highly relevant for predicting genotype-dependent mutational effects in precision medicine. Furthermore, the polymorphic essential genes discovered here should ideally be avoided as potential drug targets in future broad-spectrum antifungal drug development. This said, why mitochondrial functions and several other biological processes are enriched with essentiality polymorphism awaits future studies.

In terms of the genetic mechanisms of gene essentiality changes, we provided examples of aneuploidy-induced chromosome-wide changes of gene essentiality. Previous studies found that the essentiality change of a gene between two strains is frequently attributed to changes of multiple modifier genes[21], although simpler cases involving single modifiers are known[72]. Our study provided many examples of essentiality changes, including those of genes encoding all components of the elongator complex whose human counterpart is involved in a number of devastating neurological diseases. These cases can be analyzed in the future to identify the underlying genetic causes, a necessary step for a full understanding of the genetic background-dependency of mutational effects in general and gene essentiality in particular.

## Methods

### SNP identification and phylogenetic reconstruction.
We downloaded from NCBI the genomic sequencing data of four Chinese strains (HN8, BJ20, BJ14, and HN16) in Duan et al.[73] (BioProject identifier PRJNA396809) and 12 other strains in Maclean et al.[41] (PRJNA308843). Reads were first trimmed using Cutadapt v1.18[74] to remove adapter sequences. Burrows-Wheeler Aligner 0.7.17[75] was used to map reads to the S288C reference genome (version R64-2-1) with standard parameters. SAMTools v1.8 was employed to convert the alignment results to the BAM format[76], and Picard tools (http://broadinstitute.github.io/picard/) were used to remove duplicate sequences. Paired reads were filtered using SAMTools with parameters -f 3 -F 4 -F 8 -F 256 -F 1024 -F 2048 -q 30. SNPs were called using the Genome Analysis Toolkit (GATK) platform[77]. At any site in any strain, a variant was considered if it was supported by >5 reads and >10% of reads; otherwise, the site was considered homozygous. Finally, a total of 229,705 SNP sites were extracted from the 16 strains, and 179,416 of them were homozygous in every strain. We reconstructed a maximum composite likelihood neighbor-joining tree by MEGA7[78] using these 179,416 SNPs. To assess the strength of support for the tree, we performed 1000 bootstrap replications. Phylogenetic patterns obtained were consistent with previous reports[41,73].

### Construction of mutagenesis libraries.
Plasmid pBK257 was obtained from the Kornmann lab[25]. All yeast strains used are listed in Supplementary Data 1. The endogenous URA3 and ADE2 genes from these strains were replaced with KanMX4 and HphMX, respectively. The plasmid was transformed into each Δade2 Δura3 strain followed by selection on plates with SD medium minus uracil (SD − uracil). We collected about 5000 colonies per strain. The transformants were scraped from plates, pooled, and inoculated to 2000 ml SD − uracil + 0.2% glucose + 2% raffinose culture at an optimal density (OD) of 0.15 to enable a smooth transition during the diauxic shift from glucose to galactose. After overnight growth to saturation at 30 °C, cells were spun for 5 min, followed by the removal of 90% of supernatant. The resuspension (200 μl) was evenly plated on about 300 SD + galactose − adenine plates. Additionally, we plated 200 μl resuspension on a few SD + glucose − adenine plates to control for transposition that occurred before exposure to galactose. Plates were incubated for 20 days at 30 °C to induce transposition events. Colonies in which transposon excision repaired ADE2 started to appear in about 10 days. After 20 days, galactose plates had 100–200 colonies/cm², while each glucose plate had only a few (<10) colonies. All colonies were then scraped off the galactose plates using sterile ddH₂O, pooled, washed, and inoculated to 2000 ml SD + glucose − adenine culture at a density of $2 \times 10^6$ cells/ml. This regrowth step was used to dilute the remaining untransposed cells and dead cells. The culture was grown to saturation overnight at 30 °C and then harvested for genomic DNA extraction.

### Sequencing library preparation.
The following transposon sequencing steps were performed as described previously[25]. For each strain, genomic DNA was extracted from around $10^9$ yeast cells using a MasterPure Yeast DNA Purification Kit (Lucigen; MPY80200). Genomic DNA ($2 \times 2$ μg) was separately digested with 50 units of DpnII (NEB #R0543L) and NlaIII (NEB #R0125L). Each of these four-cutter restriction enzymes has one cutting site approximately every 256 nucleotides, and the use of two enzymes increased the probability of cleavage of the flanking sites of an inserted transposon. Following heat inactivation of the restriction enzymes at 65 °C for 20 min, the DNA fragment was circularized and ligated by 25 Weiss units of T4 Ligase (Thermo Scientific #EL0011) at 22 °C for 6 h. The DNA was precipitated overnight at −20 °C by adding 3 M sodium acetate (to a final concentration of 0.3 M), 2.5 volumes of 100% ethanol, and 5 μg linear acrylamide (Ambion AM9520). The reaction was then centrifuged at 4 °C and the supernatant was removed, followed by washing with 1 ml 70% ethanol and drying at room temperature. The DNA pellet was dissolved into 1 ml ddH₂O.

Transposon/chromosomal junction regions were enriched using PCR with the forward primer P5_MiniDs (AATGATACGGCGACCACCGAGATCTACACtcc gtcccgcaagttaaata) and reverse primer P7_MiniDs (CAAGCAGAAGACGGC ATACGAGATNNNNNNNNNacgaaaacgaacgggataaa). The two primers each contain two parts: the Illumina adaptor (uppercase letters) and a fragment from MiniDs (lowercase letters). The 8-nucleotide index in P7_MiniDs is used to distinguish among different libraries. The PCR reactions were purified using QIAquick PCR purification kit (Qiagen #29106). Equal amounts of DpnII-digested and NlaIII-digested libraries were pooled and sequenced using Illumina NextSeq 500 with a single-end 75-nucleotide strategy by two custom sequencing primers. Custom_P1 (tttaccgaccgttaccgaccgttttcatcccta) is for sequencing the flanking regions of MiniDs, while custom_P2 (GGTTTTCGATTACCGTATTTATCC CGTTCGTTTTCGT) is for reading the 8-nucleotide index in the P7_MiniDs primer.

### Sequence analysis.
The fastq file was processed with Cutadapt v1.18[74] to trim sequences with the recognition sites "GATC" for the DpnII library and "CATG" for the NlaIII library, respectively. The remaining reads were mapped to the S. cerevisiae reference genome (version R64-2-1) by Burrows-Wheeler Aligner 0.7.17 with standard parameters[75]. Aligned reads were processed and sorted using SAMtools v1.8 with the parameter set to -q 30 to filter out reads with low mapping quality[76], and the resulting bam file was transformed to bed file by bedTools bamtobed[79]. Reads of the same orientation that were mapped within two nucleotides were considered to have originated from the same transposon. All bed files were then uploaded to the UCSC genome browser.

The gene deletion-based essentiality annotation was downloaded from Saccharomyces Genome Deletion Project[39,80]. Transposon density per gene was calculated by dividing the number of transposons in the coding region of a focal gene by the effective length of the gene. All lengths refer to coding sequence lengths in nucleotide unless otherwise noted. The effective length equals the length in the reference genome minus the total length of repetitive regions minus the total length of unmapped regions from the whole-genome sequencing data. To determine the repetitive sequences in the genome, we generated simulated reads by sliding a 75-nucleotide window, with a step size of 1 nucleotide, along each chromosome of the reference genome, and then aligned it as described above. Consecutive regions with reading alignments of mapping quality below 30 were considered "repetitive sequences" because of the low mapping quality for multiple mapped reads. Unmapped regions from whole-genome sequencing data may arise from sequence divergence between the strain and the S288C reference genome or loss of the region. To identify unmapped regions, we first downloaded the whole genome sequencing data of the 16 strains from previous studies. We then mapped the fastq file to the S. cerevisiae reference genome, resulting in an average coverage of 60× per genome. We further extracted the genomic regions with sequencing coverage lower than 5 and considered them unmapped regions. Genes that had effective gene lengths lower than 300 nucleotides were discarded.

### Gene essentiality prediction.
For each gene in S288C, we considered nine features (Supplementary Data 3) in machine learning. RF[81] classification was performed using the R package "caret" with a 10-fold cross validation[82]. We treated gene deletion-based essentiality annotations in S288C (downloaded from Saccharomyces Genome Deletion Project[39] on March 9, 2019) as the ground truth. To reduce the impact of misannotated gene essentiality on the machine learning model, we built the training and testing sets by removing the following genes: (1) annotated nonessential but found essential in Michel et al.[25], (2) annotated nonessential but null mutations cause inviability according to YeastMine[83], (3) overlapped with essential genes in coding sequences, and (4) annotated auxotrophic genes in Saccharomyces Genome Database[84]. The importance of a given feature in the RF classifier was evaluated by the sum of the reduction in error when each feature is added to the model, using the function "varImp" from the R package "caster". The receiver operating characteristic (ROC) curve, which illustrates the diagnostic ability of a binary classifier by plotting the true positive rate versus the false positive rate at various thresholds of classification[85], was used to evaluate the performance of the classifier. The R package "pROC" was used to generate the ROC curve and

estimate the AUC, the area under the ROC curve[86]. When the RF output is between two preset thresholds, gene essentiality prediction is less reliable so the essentiality is considered undermined. See "Results" on the determination of the two thresholds.

Because of the lack of gene deletion-based essentiality annotations for the 15 non-S288C strains, we used the machine learning models and thresholds determined in S288C adjusted for the difference in the number of transposons between the focal strain and S288C. Specifically, because in our data S288C has the highest number of transposons in its coding regions among the 16 strains, we randomly downsampled the transposons in the coding regions of S288C to the number observed in each of the other 15 strains. For each focal strain, we used the mean features from 100 independent downsamplings as observed features to train the machine learning model and determine the thresholds. As in S288C, in each of the other 15 strains, we designated the essentiality of a gene undetermined if the prediction is between the two thresholds used to define essential and nonessential genes.

**Protein and genetic interactions**. The list of protein and genetic interactions in *S. cerevisiae* (S288C) was obtained from Biological General Repository for Interaction Datasets[87] (BioGRID) (https://www.thebiogrid.org), version 4.4.198, which contained 176,516 protein-protein and 584,211 genetic interactions, respectively.

**Rate of gene essentiality changes**. To estimate the rate of gene essentiality changes along the 16-strain phylogeny, we first used the function "hsp_max_-parsimony" from the R package "caster" (version 1.6.1) to infer the gene essentiality by parsimony[88] for strains in which the essentiality is undermined. We then used the function "asr_max_parsimony" from the same package to reconstruct the ancestral states of gene essentiality in all interior nodes of the phylogeny, followed by extracting the output "total_cost", which represents the total number of essentiality changes in the tree under parsimony. Using parsimony to infer gene essentiality and count the number of essentiality changes made our results on rate variation among genes conservative. The mean rate of gene essentiality changes is the total number of essentiality changes for all genes divided by the number of genes considered (5814). To control the impact of introgression, we randomly sampled the same number of SNPs (567) as the number of genes exhibiting essentiality polymorphism, requiring that the allele frequencies of the sampled SNPs match the essentiality frequencies of these 567 genes in the 16 strains (upon the parsimony inferences in strains with undetermined essentiality). The number of nucleotide changes per SNP was estimated by the method used for estimating the number of essentiality changes per gene.

**Evolutionary correlations and GO analysis**. Because of the phylogenetic non-independence among the 16 *S. cerevisiae* strains, we employed phylogenetically independent contrasts to estimate the evolutionary correlation in essentiality between genes across strains. A strain is removed from the analysis of a gene when its essentiality in the strain is undermined. We used the function "threshBayes" from the package "phytools" to conduct the analysis[89]. Specifically, this function uses Felsenstein's threshold model to measure the correlation between discrete characters, where the categorical states arise by thresholds on underlying continuous characters[50]. Based on the resulting correlation matrix, we conducted a hierarchical clustering with complete linkage using the heatmap.2 function in the "gplots" package[90]. We then used the program GO Term Finder[91], available from the *Saccharomyces* Genome Database[84] (https://www.yeastgenome.org), to identify functional categories enriched in each gene group defined under the cutoff of height = 6, 7, or 8. *P*-values were adjusted using the Bonferroni correction for multiple testing.

**Protein complexes and metabolic pathways**. The list of genes that encode members of each protein complex in *S. cerevisiae* (S288C) was obtained from Complex Portal Database[92] (https://www.ebi.ac.uk/complexportal/home). The list of genes that encode components of each metabolic pathway in *S. cerevisiae* (S288C) was obtained from the KEGG database[93] (https://www.genome.jp/kegg/).

**Reporting summary**. Further information on research design is available in the Nature Research Reporting Summary linked to this article.

## Data availability

The Illumina sequencing data have been deposited to NCBI SRA under the accession number PRJNA776744, and the map of transposon insertions in the 16 yeast strains is available at http://genome-euro.ucsc.edu/s/Piaopiao/samples_16strains. All other data are presented in the paper and associated supplementary materials. Source data are provided with this paper.

## Code availability

Custom code[94] is available at https://github.com/PiaopiaoChen/Gene_essentiality.git and https://doi.org/10.5281/zenodo.5907088.

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

## Acknowledgements

We are grateful to Benoît Kornmann for technical assistance with SATAY. We thank Bingyu Yan and members of the Zhang laboratory for their valuable comments. This work was supported by the U.S. National Institutes of Health research grant R35GM139484 to J.Z.

## Author contributions

P.C. and J.Z. designed the study and wrote the paper. P.C. and A.H.M. performed the experiments. P.C. analyzed the data.

## Competing interests

The authors declare no competing interests.
