## [Peer Review File · Nature Communications]

Title: Transposon insertional mutagenesis of diverse yeast strains suggests coordinated gene essentiality polymorphismsREVIEWER COMMENTS

Reviewer #1 (Remarks to the Author):

The manuscript "Transposon insertional mutagenesis of diverse yeast strains reveals coordinated gene essentiality polymorphisms" by Professor Zhang and colleagues, is a nice example of how the SATAY AcDs system, which was developed for genome-wide analysis of transposon insertions in *S. cerevisiae* by Michel, Kornmann and colleagues, can be leveraged to learn about gene functions in *S. cerevisiae*. In this paper, they study gene essentiality polymorphism across 16 phylogenetically diverse strains of *S. cerevisiae*, using SATAY for in vivo transposon mutagenesis. The study reveals that 9.2% of *S. cerevisiae* genes display an essentiality polymorphism. Essentiality polymorphism is presumably defined as a situation where the same gene is differentially essential in different strain backgrounds. Essentiality is not always a clear decision (because some essential genes still carry insertions in some domains, often near the C-terminal coding sequence) use straightforward models individually and in combination to drive the decision. While the idea of using SATAY to compare gene essentiality across different wild-type isolates is clearly an effective approach, and a logical extension of Michel et al. 2017 ELife, this is essentially an extension of that work applied to 16 isolates within the species. Several issues that need to be addressed are listed below.

There are several issues that should be addressed:

1. The terms 'essentiality polymorphism' or 'polymorphic essentiality' vs 'essentiality monomorphism' or 'monomorphic essentiality' are not defined explicitly and is not really ideal. Polymorphism means different shapes or forms, while here the point is that in different genetic backgrounds, the same gene may or may not be essential. This is more of an issue of heterogeneity in the response of the genetic background to the loss of function of a single gene (Hou 2018 ncbi.nlm.nih.gov/pmc/articles/PMC6085889/.) There must be a more appropriate term for this, as polymorphism invokes the idea of different alleles at one locus and rather than that the phenotypic outcome of the same allele is different in different genetic backgrounds.
2. When mentioning statistically significant differences, the pvalue or other parameter should be provided in the text (e.g., lines 326-326).
3. Further explanation is required regarding the models: How was the training done? Were all the genes annotated for their essentiality used in the training stage of the models ('ground truth' set)? How did the authors treated the imbalance between the number of genes annotated as essential vs. non-essential in the available 'ground truth' set?
4. As ROC curves are sensitive to class imbalance (the number of genes annotated as essential vs. non-essential in the available 'ground truth' set), these AUC values might not be reliable, unless the imbalance was addressed somehow- for example evaluating on equal numbers of randomly selected genes of each class.

5. Figure 3. Why would one expect the 'polymorphic' genes be expected to have 'intermediate' numbers of protein or genetic interactions? It appears that while this statistic may be significant, it does not appear to be useful for predicting essentiality or 'polymorphic genes' (a really confusing term in Fig 3, because polymorphic genes are classically defined as genes with multiple allele).
6. Lines 324-332—In addition to noting that metabolic pathways are over-represented in the data, it would be good to discuss why this might be.
7. Lines 369-372: this speculation is accompanied by no evidence and at the very least, an example should be tested to see if there is anything to this idea.
8. Lines 511-517: Was the pROC package used to generate the ROC curve itself or also to predict essentiality? And if so, how was it done?
9. Lines 521-523: yes, removing problematic samples will reduce the prediction errors, but isn't it true to all models and all predictions

10. The observation that most essential genes tolerate insertions is not novel and was noted in Michel et al. and thus this section should highlight what is uniquely found here that was not shown in the prior studies.

11. Line 114: 'nearly saturated transposon insertion library' is an overstatement, given the fact that there are a bit over 320,000 unique insertions per strain. Clearly this is plenty of mutants. But given that Ac/Ds can insert anywhere in the genome, true saturation library would be >12M insertions. Here there is an average of 1 insertion per 37 nucleotides, but is not 'near saturated'.

12. Other studies of Ac/Ds insertion used machine learning algorithms that measured very similar or identical features (number of insertions/gene length and number of insertions within vs flanking the ORF, to those used here (e.g., Segal et al., 2018 mBio and Levitan et al., 2020. Curr Genetics).

13. Line 158: A model that includes the 4 listed features (or at least the two relative ones) would be likely to give much more robust data without requiring that such a large proportion of genes (1682, which is just under 30% of the genes!) to be 'disqualified' for not fulfilling all 4 criteria.

14. The observation of insertions at the 3' and 5' ends of essential genes was seen in the original SATAY paper (Michel 2017 Elife) and in that paper, there were several in depth analyses showing that these domains were not essential despite the essentiality of the entire ORF. This is worth citing.

15. Line 222—it is not clear to this reviewer why a Poisson distribution is expected and this should be stated more explicitly. Are the underlying assumptions robust?

16. Lines 303-308—do these differentially essential genes have any role in the nutrients available to the clinical strains (in a human host) vs the nutrients available to industrial or environmental yeasts?

17. Figure 4. Why is the mitochondrial matrix treated differently from the other three mitochondrial GO

terms? What are the biological implications of the GO terms and pathways that were identified here? Could some aspect of the phenotypic differences between these strains provide a biological context to the results?

Reviewer #2 (Remarks to the Author):

Thank you for the opportunity to review “Transposon insertional mutagenesis of diverse yeast strains reveals coordinated gene essentiality polymorphisms” by Chen and Zhang. In this study authors investigate conditional essential genes across 16 yeast strains. Authors use transposon mutagenesis and build a model using S288c known essential genes to predict essential genes in the other 15 yeast strains. They show that genes that show essentiality switches do not follow the phylogeny of the 16 yeast strains. They show that polymorphic essential genes, those that switch essentiality status across strains, have protein-protein interaction and genetic interaction degree in between monomorphic essential genes and monomorphic nonessential genes. They show that genes in certain bioprocesses, protein complexes and pathways have similar profiles of essentiality status across 16 strains. By dividing each gene into 10 segments, authors identify essential ‘gene parts’. Authors suggest that the knowledge of polymorphic genes in yeast is important for devising antifungals and provide insight into precision medicine approaches.

The idea to characterize the essential gene set across 16 natural yeast isolates is important and fills a gap in knowledges in the field. The manuscript is generally well-written although I have some comments.

Main points

1. The introduction never mentions the term ‘missing heritability’, which is essentially what authors claim that they give insight to with this study and its relation to human health. Authors should mention this and a relevant citation Zuk et al PNAS 2012 comes to mind. Additionally, a mention of context specific essentiality in different cancer subtypes is also important here and authors should cite the DepMap studies along with others such as Wang et al Science 2015, Hart et al Cell 2015, Boone and Andrews Science 2015 etc.
2. Authors cite Dowell et al Science 2010 study on conditional essential genes, but they should include a more recent study from the same lab: Hou et al PNAS 2019.
3. Does the peak on the left in Fig. 1e represent enrichment of transposon insertions in the centromere? This should be clearly stated in the text about this particular figure. What other genes fall in this peak? Does targeting of the centromere by transposons induce gross chromosomal abnormalities resulting in aneuploidies that authors detected in this study? This should be discussed in the main text.
4. Method 4 for essential gene prediction is clearly more superior to other methods. So why do authors proceed with predictions under all 4 models? Also, why restrict to the 4,621 genes. Authors should provide the prediction for the rest of the genes under Model 4.
5. The statement on lines 176-178 is confusing. Is the minimal effect predicted to be because of the

small number of genes that show the disparity between prediction and ground truth? Along the same lines, where did you obtain the ground truth? Is it through Yeast Mine? Provide the date of accession of the database and exactly what was retrieved.

6. Figure numbers should be provided directly on the figure panels which would help the reviewers to navigate the paper. They should be included in the revised manuscript.

7. The section 'Protein/genetic interactions and polymorphic gene essentiality' is out place. Its natural place is after the section entitled 'Correlated essentiality changes among genes'. Since network degree correlate with gene age, authors should discuss this property in relation to essentiality switches.

8. Authors should explicitly define what they mean by monomorphic and polymorphic essential/nonessential genes.

9. Line 259 discusses degree differences for essential and nonessential genes on the digenic interaction network. This property was also shown to be true for the trigenic interaction network Kuzmin et al Science 2018. This should be included.

10. Authors include in the conclusions the discussion of van Leeuwen et al Molecular Systems Biology 2020. Authors should do a formal comparison with these data to see if suppressors of essential genes explain their essentiality switches.

11. Data S8 should be presented as a figure to improve clarity of the manuscript.

12. The statement in lines 295-296 about mitochondria should be explained. Why is this the case?

13. Line 324-325 discusses an analysis but shows no corresponding figure or datafile. This should be explained. How was this analysis done? What other processes were tested?

14. The statement on line 369-372 is confusing. Are you referring to alternative splice variants or to a modular gene product with different domains some of which are essential and others are nonessential for the protein function? Can you provide an example from the literature which is consistent with your data?

15. In the supplementary figure 5 why is the genetic interaction degree of polymorphic essential genes less than the genetic interaction degree of polymorphic nonessential genes? What about the PPI network?

Minor point

Every figure legend should explicitly state what is shown in red and blue and what do the white arrows in the red/blue boxes mean. I am assuming it is the directionality of a given gene but this is not stated.

Reviewer #3 (Remarks to the Author):

The underlying determinants of being an essential gene in a particular organism have been one of the central questions in cell and evolutionary biology. In this study, Chen and Zhang investigated the polymorphisms in gene essentiality among diverse yeast strains in an effort to shed light on the property and evolution of gene essentiality. They observed associations between the change in gene essentiality and the number of protein-protein or genetic interactions. They further found that gene essentiality changes tended to occur concordantly among subunits of the same protein complex, among the

components of the same metabolic pathway, or among proteins that share the same subcellular localization. I think it is a smart idea to use transposon insertional mutagenesis to address this very question in a high-throughput manner. Overall the analyses have been carefully conducted, the conclusions are supported by the data, and the paper is well written. My detailed comments are listed below.

1. I wonder if there are any biases in locations of miniDs insertions and detection (using DpnII digestion with reverse PCR). The paper has discussed some tendencies of transposon insertion. Still, I think it is worth a bit more investigation as it constitutes the null model for this particular study about essentiality.
2. The authors use the parsimony principle to infer the number of historical events in the evolution that the essentiality has changed. In addition to the possibility of introgression that the authors have discussed, I also have some concerns about if the parsimony principle can be used here. As shown in Fig. 2c, many changing events in essentiality did not align with the phylogenetic tree. This is not unexpected as there are numerous mechanisms for the change in gene essentiality. Therefore, I would suggest the authors also perform the analyses in Fig. 2d and 2e simply based on the number of strains the essentiality has changed for a gene.
3. For the “central part essential” and “multiple discontinuous parts essential” examples presented in Fig. 5d, I wonder if some essential non-coding transcripts exist in these regions, causing the indispensable regions. The host genes for these transcripts are actually not essential. Can the authors check strand-specific RNA-seq data for some potential signals along this vein?

Response to reviewers

We are grateful to the three reviewers for their constructive comments, which have helped improve our manuscript significantly. Below please find our point-to-point response in blue.

Reviewer: #1

The manuscript “Transposon insertional mutagenesis of diverse yeast strains reveals coordinated gene essentiality polymorphisms” by Professor Zhang and colleagues, is a nice example of how the SATAY AcDs system, which was developed for genome-wide analysis of transposon insertions in *S. cerevisiae* by Michel, Kornmann and colleagues, can be leveraged to learn about gene functions in *S. cerevisiae*. In this paper, they study gene essentiality polymorphism across 16 phylogenetically diverse strains of *S. cerevisiae*, using SATAY for in vivo transposon mutagenesis. The study reveals that 9.2% of *S. cerevisiae* genes display an essentiality polymorphism. Essentiality polymorphism is presumably defined as a situation where the same gene is differentially essential in different strain backgrounds. Essentiality is not always a clear decision (because some essential genes still carry insertions in some domains, often near the C-terminal coding sequence) use straightforward models individually and in combination to drive the decision. While the idea of using SATAY to compare gene essentiality across different wild-type isolates is clearly an effective approach, and a logical extension of Michel et al. 2017 ELife, this is essentially an extension of that work applied to 16 isolates within the species. Several issues that need to be addressed are listed below.

There are several issues that should be addressed:

1. The terms ‘essentiality polymorphism’ or ‘polymorphic essentiality’ vs ‘essentiality monomorphism’ or ‘monomorphic essentiality’ are not defined explicitly and is not really ideal. Polymorphism means different shapes or forms, while here the point is that in different genetic backgrounds, the same gene may or may not be essential. This is more of an issue of heterogeneity in the response of the genetic background to the loss of function of a single gene (Hou 2018.) There must be a more appropriate term for this, as polymorphism invokes the idea of different alleles at one locus and rather than that the phenotypic outcome of the same allele is different in different genetic backgrounds.

We appreciate this comment on terminology. Polymorphism means genotypic or phenotypic variation among individuals of the same species. Here, we use essentiality polymorphism to refer to the phenotypic variation (live or death) among individuals carrying transposon insertions in the same gene, so the use of polymorphism is not wrong. We completely agree that we are studying “heterogeneity in the response of the genetic background to the loss of function of a single gene”, but this is too long of a phrase. If the reviewer has a better suggestion that is less wordy, we would be happy to consider. For the time being, we still use “essentiality polymorphism” and have added a definition to the main text (page 4, end of paragraph 1). We also defined polymorphic essential genes and monomorphic essential/nonessential genes in the main text (page 9, paragraph 2) as well as in the legend of Fig. 3.

2. When mentioning statistically significant differences, the pvalue or other parameter should be provided in the text (e.g., lines 326-326).

The p-values were presented in figures or figure legends in the previous version, because we felt that it would be more convenient for readers to see these p-values when viewing figures. We are under the impression that redundant information is discouraged by the journal. We would be happy to present the p-values in both the main text and figures should this redundancy be allowed.

3. Further explanation is required regarding the models: How was the training done? Were all the genes annotated for their essentiality used in the training stage of the models ('ground truth' set)? How did the authors treated the imbalance between the number of genes annotated as essential vs. non-essential in the available 'ground truth' set?

Because of the reviewer's suggestion in Comment 12, we have now adopted a machine learning algorithm to estimate gene essentiality. This comment is no longer relevant. Please see response to Comment 12 for details about the machine learning-based prediction.

4. As ROC curves are sensitive to class imbalance (the number of genes annotated as essential vs. non-essential in the available 'ground truth' set), these AUC values might not be reliable, unless the imbalance was addressed somehow- for example evaluating on equal numbers of randomly selected genes of each class.

Because of the reviewer's suggestion in Comment 12, we have now adopted a machine learning algorithm to estimate gene essentiality. This comment may no longer be relevant. In any case, Fig. S2b and Fig. S2c provide more detailed information about the prediction accuracy.

5. Figure 3. Why would one expect the 'polymorphic' genes be expected to have 'intermediate' numbers of protein or genetic interactions? It appears that while this statistic may be significant, it does not appear to be useful for predicting essentiality or 'polymorphic genes' (a really confusing term in Fig 3, because polymorphic genes are classically defined as genes with multiple allele).

A gene with polymorphic essentiality is essential in some individuals but nonessential in other individuals of the same species. As mentioned in the response to Comment 1, relevant definitions have been added to the manuscript.

Because the number of protein interaction partners of essential genes was found significantly greater than that of nonessential genes, it makes intuitive sense that the genes with polymorphic essentiality will have an intermediate number of interactions. We do not use this pattern to predict gene essentiality. Rather, we suggest that there may be a causal relationship between the number of protein or genetic interactions and gene essentiality polymorphism. This is now elaborated in the manuscript (end of page 10).

6. Lines 324-332—In addition to noting that metabolic pathways are over-represented in the data, it would be good to discuss why this might be.

We agree. We found that metabolic pathway genes and protein complex members exhibit significantly greater essentiality polymorphism than other genes. This may be due to a large number of indispensable redundancies in metabolic networks and protein complexes (Wang and Zhang, 2009 *Genome Biol Evol*; Li et al., 2010 *PLOS Genet*); mutations that remove these redundancies may have fitness effects only in some environments so can spread in other environments. Consequently, genes involved in these pathways/complexes are prone to essentiality polymorphism. This point has been added to the manuscript (page 14, paragraph 2).

7. Lines 369-372: this speculation is accompanied by no evidence and at the very least, an example should be tested to see if there is anything to this idea.

While experimentally testing the operon hypothesis is beyond the scope of the present study, we now cite a number of papers reporting operons (multiple independent proteins encoded on a single molecule of mRNA) in fungi, plants, green algae, nematodes, and flies (page 15, paragraph 2).

8. Lines 511-517: Was the pROC package used to generate the ROC curve itself or also to predict essentiality? And if so, how was it done?

Because of the reviewer's suggestion in Comment 12, we have now adopted a machine learning algorithm to estimate gene essentiality. This comment is no longer relevant. Please see response to Comment 12 for details about the machine learning-based prediction.

9. Lines 521-523: yes, removing problematic samples will reduce the prediction errors, but isn't it true to all models and all predictions

Because of the reviewer's suggestion in Comment 12, we have now adopted a machine learning algorithm to estimate gene essentiality. This comment is no longer relevant. Please see response to Comment 12 for details about the machine learning-based prediction.

10. The observation that most essential genes tolerate insertions is not novel and was noted in Michel et al. and thus this section should highlight what is uniquely found here that was not shown in the prior studies.

We agree. As we previously wrote, "several studies reported that some annotated essential genes tolerate transposon insertions in one or more segments of their coding regions". However, the prevalence of this phenomenon and the conservation of this phenomenon across *Saccharomyces cerevisiae* strains are unknown, so we focused on these questions in the present work and have clarified this point (page 14, paragraph 3).

11. Line 114: 'nearly saturated transposon insertion library' is an overstatement, given the fact that there are a bit over 320,000 unique insertions per strain. Clearly this is plenty of mutants. But given that Ac/Ds can insert anywhere in the genome, true saturation library would be >12M insertions. Here there is an average of 1 insertion per 37 nucleotides, but is not 'near saturated'.

We meant “nearly saturated at the gene level” (not at the nucleotide level), and have clarified this point (page 5, paragraph 1).

12. Other studies of AcDs insertion used machine learning algorithms that measured very similar or identical features (number of insertions/gene length and number of insertions within vs flanking the ORF, to those used here (e.g., Segal et al., 2018 mBio and Levitan et al., 2020. Curr Genetics).

Lots of thanks for this comment! We have followed the suggestion to use a machine learning algorithm to predict gene essentiality, which is more reliable than the method we previously used. Please see details in the section titled “Gene essentiality classification using machine learning” (starting on page 6) and the corresponding section in Methods (starting on page 20).

13. Line 158: A model that includes the 4 listed features (or at least the two relative ones) would be likely to give much more robust data without requiring that such a large proportion of genes (1682, which is just under 30% of the genes!) to be ‘disqualified’ for not fulfilling all 4 criteria.

Because we now use a machine learning algorithm to estimate gene essentiality, this comment is no longer relevant. Now the essentiality of 5,814 genes have been predicted.

14. The observation of insertions at the 3’ and 5’ ends of essential genes was seen in the original SATAY paper (Michel 2017 Elife) and in that paper, there were several in depth analyses showing that these domains were not essential despite the essentiality of the entire ORF. This is worth citing.

We have added that “The successful removal from *TAF3* and *PRP45* of the 3’ regions that tolerate transposon insertions supports this explanation.” (page 15, paragraph 2).

15. Line 222—it is not clear to this reviewer why a Poisson distribution is expected and this should be stated more explicitly. Are the underlying assumptions robust?

According to the probability theory, when the rate of occurrence of an event is constant, the number of events in a given period of time follows a Poisson distribution. Under the assumption that the rate of essentiality change is the same for all genes, the number of changes per gene follows a Poisson distribution. This has been clarified in the manuscript (page 9, paragraph 3).

16. Lines 303-308—do these differentially essential genes have any role in the nutrients available to the clinical strains (in a human host) vs the nutrients available to industrial or environmental yeasts?

While all six ELP complex subunits as well as the cofactor KTI12 are nonessential in the vast majority of the yeast strains surveyed, they are all essential in the clinical strain 322134S isolated from human throat/sputum. Interestingly, ELP genes are essential to laboratory strains W303 and SEY6210, which are highly closely related to S288C, at 37°C but not 30°C, suggesting the possibility of an alternative elongator that works at 30°C but not 37°C. Genes encoding this alternative elongator may be inactivated or lost in strain 322134S that lives at 37°C;

consequently, ELP genes become essential to the strain if the temperature lowers to 30°C. We have added this possibility to the manuscript (end of page 12).

17. Figure 4. Why is the mitochondrial matrix treated differently from the other three mitochondrial GO terms? What are the biological implications of the GO terms and pathways that were identified here? Could some aspect of the phenotypic differences between these strains provide a biological context to the results?

Mitochondrial matrix was treated the same as the other three mitochondrial GO terms. Genes exhibiting essentiality polymorphism are significantly enriched with mitochondrial gene expression, mitochondrial translation, and several metabolic and biosynthetic processes (Fig. S7). Regarding the essentiality changes of many genes involved in mitochondrial functions, we have added the following discussion to page 12, paragraph 1.

*“Interestingly, Kim et al. reported 95 genes related to mitochondrial function that have opposite gene essentiality between fission yeast and budding yeast¹⁹, 53 of which belong to this group. Clearly, mitochondrial functions are frequently involved in yeast intraspecific and interspecific gene essentiality changes. Notably, the loss of mitochondrial DNA is non-lethal to many *S. cerevisiae* strains, a phenomenon known as petite-positive, but the loss becomes lethal (known as petite-negative) upon the inactivation of genes encoding F1-ATPase subunits, ATP/ADP carrier, i-AAA protease complex, phosphatidylglycerophosphate synthase, or mitochondrial protein import components⁵¹⁻⁵³. It is possible that DBVPG1107, DBVPG4651, WE372, and BJ20 are petite-negative just like fission yeast.”*

This said, little is known about the underlying reasons of many other essentiality changes. More detailed genetic analysis (as in Hou et al. 2019 PNAS) should be done in the future for a full understanding of the genetic background-dependency of mutational effects in general and gene essentiality polymorphism in particular. This is now discussed in the manuscript (pages 16-17).

Reviewer: #2

Thank you for the opportunity to review “Transposon insertional mutagenesis of diverse yeast strains reveals coordinated gene essentiality polymorphisms” by Chen and Zhang. In this study authors investigate conditional essential genes across 16 yeast strains. Authors use transposon mutagenesis and build a model using S288c known essential genes to predict essential genes in the other 15 yeast strains. They show that genes that show essentiality switches do not follow the phylogeny of the 16 yeast strains. They show that polymorphic essential genes, those that switch essentiality status across strains, have protein-protein interaction and genetic interaction degree in between monomorphic essential genes and monomorphic nonessential genes. They show that genes in certain bioprocesses, protein complexes and pathways have similar profiles of essentiality status across 16 strains. By dividing each gene into 10 segments, authors identify essential ‘gene parts’. Authors suggest that the knowledge of polymorphic genes in yeast is important for devising antifungals and provide insight into precision medicine approaches.

The idea to characterize the essential gene set across 16 natural yeast isolates is important and fills a gap in knowledges in the field. The manuscript is generally well-written although I have some comments.

Main points

1. The introduction never mentions the term ‘missing heritability’, which is essentially what authors claim that they give insight to with this study and its relation to human health. Authors should mention this and a relevant citation Zuk et al PNAS 2012 comes to mind. Additionally, a mention of context specific essentiality in different cancer subtypes is also important here and authors should cite the DepMap studies along with others such as Wang et al Science 2015, Hart et al Cell 2015, Boone and Andrews Science 2015 etc.

We agree and have added this point and relevant references (page 3, paragraphs 1 and 2).

2. Authors cite Dowell et al Science 2010 study on conditional essential genes, but they should include a more recent study from the same lab: Hou et al PNAS 2019.

Hou et al. (PNAS 2019) is cited in Discussion where we discuss the mechanism of gene essentiality changes, because this paper is not about essentiality changes in general but focuses on two modifiers that contribute to the essentiality changes of two genes in the cysteine biosynthesis pathway between $\Sigma 1278b$ and S288C.

3. Does the peak on the left in Fig. 1e represent enrichment of transposon insertions in the centromere? This should be clearly stated in the text about this particular figure. What other genes fall in this peak? Does targeting of the centromere by transposons induce gross chromosomal abnormalities resulting in aneuploidies that authors detected in this study? This should be discussed in the main text.

There are two blue peaks on the left in Fig. 1e, one is near 0 (mostly essential genes misannotated as nonessential) and the other is near 0.02 on the X-axis. Neither of them is related to centromeres. Centromeres are enriched with transposon insertions while the left peaks have relatively low transposon densities. We extracted 282 genes that are located within 20,000 bp from centromeres, and found that 88.2% of these genes had a transposon density higher than 0.05 (see figure below), confirming that genes close to centromeres have a higher transposon density than other genes.

The observation that an extra copy of a chromosome renders all essential genes on the

chromosome tolerant to transposon insertions suggests that aneuploidy occurred prior to transposon insertions. We have now mentioned this point in the manuscript (page 8, paragraph 3).

4. Method 4 for essential gene prediction is clearly more superior to other methods. So why do authors proceed with predictions under all 4 models? Also, why restrict to the 4,621 genes. Authors should provide the prediction for the rest of the genes under Model 4.

Following the suggestion of reviewer 1, we now use a machine learning method to predict gene essentiality, making this comment no longer relevant. We now predict the essentiality of 5,814 genes. Please see details in the section titled “Gene essentiality classification using machine learning” (starting on page 6) and the corresponding section in Methods (starting on page 20).

5. The statement on lines 176-178 is confusing. Is the minimal effect predicted to be because of the small number of genes that show the disparity between prediction and ground truth? Along the same lines, where did you obtain the ground truth? Is it through Yeast Mine? Provide the date of accession of the database and exactly what was retrieved.

The prediction-annotation disparities should have minimal impacts because all subsequent analyses are from SATAY data, which are from the same medium and use the same method of mutagenesis (transposon insertions). The annotation is based on gene deletion in a different medium. The ground truth was obtained from *Saccharomyces* Genome Deletion Project and is presented in the column “Annotation” of Data S4. This is now clarified (page 6, paragraph 2).

6. Figure numbers should be provided directly on the figure panels which would help the reviewers to navigate the paper. They should be included in the revised manuscript.

Added as requested.

7. The section ‘Protein/genetic interactions and polymorphic gene essentiality’ is out place. Its natural place is after the section entitled ‘Correlated essentiality changes among genes’. Since network degree correlate with gene age, authors should discuss this property in relation to essentiality switches.

The section “Protein/genetic interactions and polymorphic gene essentiality” (Fig. 3) and the previous section “Gene essentiality polymorphism in yeast” (Fig. 2) are all about essentiality changes of individual genes, while the section “Correlated essentiality changes among genes” (Fig. 4) is mainly about correlated essentiality changes of multiple genes. Hence, we feel that it is more natural to talk about Fig. 4 after Fig. 3, not the other way around. The correlation between network degree and gene age is likely spurious, as multiple studies have shown that gene age estimation is unreliable (Moyers and Zhang 2015, 2016, Mol Biol Evol; Weisman et al. 2020, PLOS Biol).

8. Authors should explicitly define what they mean by monomorphic and polymorphic essential/nonessential genes.

Polymorphic essential genes are essential in some individuals but nonessential in other individuals of the same species under the same environment. By contrast, monomorphic essential genes are essential in all individuals of the same species so far examined under the same environment. We have followed the suggestion to define these two terms (page 9, paragraph 2; legend of Fig. 3).

9. Line 259 discusses degree differences for essential and nonessential genes on the digenic interaction network. This property was also shown to be true for the trigenic interaction network Kuzmin et al Science 2018. This should be included.

Added as suggested.

10. Authors include in the conclusions the discussion of van Leeuwen et al Molecular Systems Biology 2020. Authors should do a formal comparison with these data to see if suppressors of essential genes explain their essentiality switches.

van Leeuwen *et al.* reported 124 essential genes that are dispensable and subject to bypass suppression by a systematic analysis of the genetic context dependency of the essentiality of 728 essential genes in yeast. We observed that 38 essential genes in S288C can become nonessential in at least one of the 16 strains surveyed here. Among the 38 genes, 18 overlap with the dispensable essential genes observed in van Leeuwen *et al.*'s study, significantly more than the random expectation of 4 overlaps ($P < 0.0001$, permutation test). In 12 of these 18 cases, bypass suppressor genes were found, including 6 cases with loss-of-function mutations in the suppressor genes and 4 cases with gain-of-function mutations in the suppressor genes. However, by examining the SNPs and indels of the 16 strains in our study, we did not observe any loss-of-function mutations in the suppressor genes of the former 6 cases, nor did we observe the same gain-of-function mutations in the suppressor genes of the latter 4 cases. Thus, the genetic mechanisms of essentiality polymorphism in our study likely vary from those in van Leeuwen *et al.*'s study. We added the above comparison to Discussion (page 16, paragraph 2).

11. Data S8 should be presented as a figure to improve clarity of the manuscript.

We have followed the suggestion to add Fig. S7 to present the GO analysis of the 567 genes that exhibit essentiality polymorphism, and have retained Data S8 that lists the genes in each GO term.

12. The statement in lines 295-296 about mitochondria should be explained. Why is this the case?

Regarding the essentiality changes of many genes involved in mitochondrial functions, we have added the following discussion to page 12, paragraph 1.

“Interestingly, Kim et al. reported 95 genes related to mitochondrial function that have opposite gene essentiality between fission yeast and budding yeast¹⁹, 53 of which belong to this group. Clearly, mitochondrial functions are frequently involved in yeast intraspecific and interspecific gene essentiality changes. Notably, the loss of mitochondrial DNA is non-lethal to many S.

cerevisiae strains, a phenomenon known as petite-positive, but the loss becomes lethal (known as petite-negative) upon the inactivation of genes encoding F1-ATPase subunits, ATP/ADP carrier, i-AAA protease complex, phosphatidylglycerophosphate synthase, or mitochondrial protein import components⁵¹⁻⁵³. It is possible that DBVPG1107, DBVPG4651, WE372, and BJ20 are petite-negative just like fission yeast.”

13. Line 324-325 discusses an analysis but shows no corresponding figure or datafile. This should be explained. How was this analysis done? What other processes were tested?

The corresponding figure is Fig. 4i. We downloaded the metabolic pathway genes from the KEGG database (described in Methods on page 22), followed by computing the proportion of genes exhibiting essentiality polymorphism among genes that encode metabolic pathway components and the corresponding proportion among genes that do not encode metabolic pathway components. We found the former higher than the latter ($P = 7.07 \times 10^{-6}$, chi-squared test; Fig. 4i legend). We did not test other processes.

14. The statement on line 369-372 is confusing. Are you referring to alternative splice variants or to a modular gene product with different domains some of which are essential and others are nonessential for the protein function? Can you provide an example from the literature which is consistent with your data?

Yeast has almost no alternative splicing. The scenario of “different domains some of which are essential and others are nonessential for the protein function” does not explain the observation that some genes have multiple discontinuous transposon-free regions. Because *MiniDs* transposon contains multiple stop codons in each reading frame, all positions upstream of the last essential domain should be transposon-free.

We suspect that polycistronic mRNAs may be involved here. That is, multiple separate proteins are encoded on a single mRNA. Polycistronic mRNAs are common in prokaryotes. Over the past decade, a growing number of polycistronic mRNAs have been identified in eukaryotes including fungi (Gordon et al., 2015 PLoS One), plants (Wang et al., 2019 Nat. Commun, García-Ríos et al., 1997 PNAS), green algae (Gallaher et al., 2021 PNAS), nematodes (Evans et al., PNAS), and flies (Crosby et al., 2015 G3). This said, future studies are needed to test this possibility in yeast. The above information has been added to the manuscript (page 15, paragraph 2).

15. In the supplementary figure 5 why is the genetic interaction degree of polymorphic essential genes less than the genetic interaction degree of polymorphic nonessential genes? What about the PPI network?

Genetic interactions are identified through gene deletions. Because essential genes cannot be deleted, genetic interactions are studied differently for essential and nonessential genes in S288C. Consequently, their numbers of genetic interactions are not directly comparable. However, probing PPIs does not involve gene deletion, so essential and nonessential genes can be directly compared for their numbers of PPIs. We have clarified this reason in the manuscript (page 11, paragraph 2). Note that, by definition, polymorphic essential genes are also

polymorphic nonessential genes. The word essential/nonessential here refers to the status in the reference strain (S288C).

Minor point

Every figure legend should explicitly state what is shown in red and blue and what do the white arrows in the red/blue boxes mean. I am assuming it is the directionality of a given gene but this is not stated.

Yes, the white arrow in each red/blue box represents the 5' to 3' direction of the gene. We have followed the suggestion to clarify the meanings of the colors and arrows.

Reviewer: #3

The underlying determinants of being an essential gene in a particular organism have been one of the central questions in cell and evolutionary biology. In this study, Chen and Zhang investigated the polymorphisms in gene essentiality among diverse yeast strains in an effort to shed light on the property and evolution of gene essentiality. They observed associations between the change in gene essentiality and the number of protein-protein or genetic interactions. They further found that gene essentiality changes tended to occur concordantly among subunits of the same protein complex, among the components of the same metabolic pathway, or among proteins that share the same subcellular localization. I think it is a smart idea to use transposon insertional mutagenesis to address this very question in a high-throughput manner. Overall the analyses have been carefully conducted, the conclusions are supported by the data, and the paper is well written. My detailed comments are listed below.

1. I wonder if there are any biases in locations of miniDs insertions and detection (using DpnII digestion with reverse PCR). The paper has discussed some tendencies of transposon insertion. Still, I think it is worth a bit more investigation as it constitutes the null model for this particular study about essentiality.

Regarding insertion biases, Ac/Ds transposons tend to insert near the donor site (e.g., centromeres and the *ADE2* gene) and in nucleosome-free genomic regions (Michel et al. 2017 eLife; Vollbrecht et al. 2010 Plant Cell). These two biases are indeed observed here (Fig. S1ab). Ac/Ds transposons do not have a strong insertion site preference in maize or in other organisms such as *S. cerevisiae* and insert throughout the genome (Michel et al. 2017 eLife; Mielich et al. 2018 G3; Lazarow et al. 2012 Genetics; Levitan et al. 2020 Curr Genet) (now mentioned on page 5, paragraph 1). With the exception of telomere repetitive regions, no large genomic regions (>10kb) appear transposon-free in our data (Fig. S1c).

To reduce the detection biases resulting from digestion, we used two four-cutter restriction enzymes (DpnII and NlaIII) to construct two separate libraries for each strain, increasing the probability of cleavage of the flanking sites of an inserted transposon. Because the efficiency of circularization may be relatively low for long DNA fragments, we computationally examined the distribution of the expected fragment size after enzyme digestion. We found that, on average, one cleavage site exists every 166 nucleotides in the yeast genome (see figure below). Only 202 fragments are > 1 kb, and the longest is 2.3 kb. Therefore, the detection bias is limited.

2. The authors use the parsimony principle to infer the number of historical events in the evolution that the essentiality has changed. In addition to the possibility of introgression that the authors have discussed, I also have some concerns about if the parsimony principle can be used here. As shown in Fig. 2c, many changing events in essentiality did not align with the phylogenetic tree. This is not unexpected as there are numerous mechanisms for the change in gene essentiality. Therefore, I would suggest the authors also perform the analyses in Fig. 2d and 2e simply based on the number of strains the essentiality has changed for a gene.

The suggestion is equivalent to estimating the number of essentiality changes under a star phylogeny of the strains. As can be seen in the figure below, the result obtained is similar to Fig. 2d. This is now mentioned in the manuscript (page 10, paragraph 1).

In Fig. 2e, we randomly sampled the same number of SNPs as the number of genes exhibiting essentiality polymorphism, requiring that the allele frequencies of the sampled SNPs match the essentiality frequencies of these 567 genes in the 16 strains. The phylogeny, regardless of its exact shape, has been controlled in the comparison in Fig. 2e.

3. For the “central part essential” and “multiple discontinuous parts essential” examples presented in Fig. 5d, I wonder if some essential non-coding transcripts exist in these regions, causing the indispensable regions. The host genes for these transcripts are actually not essential. Can the authors check strand-specific RNA-seq data for some potential signals along this vein?

We agree that the presence of multiple essential non-coding transcripts could be another explanation of “multiple discontinuous parts essential”, so have added it in the manuscript (page 15, paragraph 2). However, this hypothesis cannot be tested using RNA-seq data, because the RNA-seq data do not tell whether an RNA is essential or not.

REVIEWER COMMENTS

Reviewer #1 (Remarks to the Author):

The manuscript is now clearer and the application of the machine learning approach clearly improves it. The discussion regarding the variability of essentiality is interesting and is now computationally justified (lines 244-285).

A few points for the authors to consider.

1. I think the term 'essentiality variation' is much more intuitive than 'essentiality polymorphism' and suggest reconsidering this terminology. Instead of ess. Monomorphism I would say "essentiality uniformity" or "essentiality constancy", which are more clearly understood.

2. End of page 4: "SATAY creates millions of cells each with an independent transposon insertion into the genome."

This is largely the case, but it is worth noting that jackpot events arise due to transposition events that can occur before induction of transposition. In such cases, not all of the transposition events (individual colonies collected) are independent.

3. Point #15. It is not clear to this reviewer that the "rate of essentiality change is the same for all genes"—for example, genes involved in metabolism may have much lower rates than for genes involved in rarely expressed structural proteins.

Reviewer #2 (Remarks to the Author):

Most comments have been addressed. However, an important point was not adequately addressed:

In the supplementary figure 6 the genetic interaction degree of polymorphic essential genes less than the genetic interaction degree of polymorphic nonessential genes. Temperature sensitive alleles of essential genes are typically used for genetic interaction studies of essential genes. Even though ts mutants are point mutants and not deletion mutants as correctly pointed out by the authors of this manuscript, they can certainly be compared as both are LOF as was done in numerous other studies and in line with what the authors are attempting to do in this manuscript. It is not clear to me how the difference in deletion vs point mutation status of a mutant affects the polymorphic genes whereby ES are lower in genetic interaction degree than NES but doesn't affect monomorphic genes whereby ES are

higher in genetic interaction degree than NES. Rather than mutation type (deletion vs point mutation of a ts mutant) the more likely explanation is that there is a bias of gene annotation in the polymorphic NES genes to bioprocesses with higher degree genes. It is true that all polymorphic genes can be ES and NES in different strains but the authors of this manuscript conduct a genetic interaction degree analysis using S288C data so the ES and NES split should mirror what they are seeing for monomorphic genes and it doesn't. This needs to be explained.

Also, authors' statement "Note that the NPI's of essential and nonessential genes are comparable because measuring protein interactions does not involve deleting genes." on page 11 par 2 is nonsensical and needs to be revised. Is it true only for polymorphic ES vs NES genes or between polymorphic and monomorphic genes or between monomorphic ES vs NES genes. Generally essential genes have more protein interactions than nonessential genes since they are hubs on the genetic and protein interaction networks. If the PPI network does mirror the GI network, this needs to be discussed formally and the associated analysis should be provided.

Reviewer #3 (Remarks to the Author):

The authors have satisfactorily addressed my previous concerns and I am delighted to recommend the publication of the study in Nature communications.

Response to reviewers

We are grateful for the additional comments from the reviewers. Below please find our point-to-point response in blue.

Reviewer #1:

The manuscript is now clearer, and the application of the machine learning approach clearly improves it. The discussion regarding the variability of essentiality is interesting and is now computationally justified (lines 244-285).

A few points for the authors to consider.

1. I think the term 'essentiality variation' is much more intuitive than 'essentiality polymorphism' and suggest reconsidering this terminology. Instead of ess. monomorphism I would say “essentiality uniformity” or “essentiality constancy”, which are more clearly understood.

We thank the reviewer for the suggested terms. Polymorphism and monomorphism respectively refer to the presence and absence of *intraspecific* variations, while variation and uniformity/constancy do not differentiate within-species from between-species comparisons. Because our study is specifically about within-species variation in gene essentiality, we feel that the use of polymorphism and monomorphism is more precise. The other reviewers are also fine with these terms.

2. End of page 4: “SATAY creates millions of cells each with an independent transposon insertion into the genome.” This is largely the case, but it is worth noting that jackpot events arise due to transposition events that can occur before induction of transposition. In such cases, not all of the transposition events (individual colonies collected) are independent.

We agree, and have revised the sentence to “SATAY creates millions of cells each with an independent transposon insertion into the genome (except in the rare instance of transposition prior to induction).” (line 94)

3. Point #15. It is not clear to this reviewer that the “rate of essentiality change is the same for all genes”—for example, genes involved in metabolism may have much lower rates than for genes involved in rarely expressed structural proteins.

The quoted sentence is our null hypothesis. We tested this hypothesis and found it to be false (see Fig. 2d). Hence, our finding supports the reviewer’s view (see the sentence starting line 281).

Reviewer #2:

Most comments have been addressed. However, an important point was not adequately addressed:

In the supplementary figure 6 the genetic interaction degree of polymorphic essential genes less than the genetic interaction degree of polymorphic nonessential genes. Temperature sensitive alleles of essential genes are typically used for genetic interaction studies of essential genes.

Even though ts mutants are point mutants and not deletion mutants as correctly pointed out by the authors of this manuscript, they can certainly be compared as both are LOF as was done in numerous other studies and in line with what the authors are attempting to do in this manuscript. It is not clear to me how the difference in deletion vs point mutation status of a mutant affects the polymorphic genes whereby ES are lower in genetic interaction degree than NES but doesn't affect monomorphic genes whereby ES are higher in genetic interaction degree than NES. Rather than mutation type (deletion vs point mutation of a ts mutant) the more likely explanation is that there is a bias of gene annotation in the polymorphic NES genes to bioprocesses with higher degree genes. It is true that all polymorphic genes can be ES and NES in different strains but the authors of this manuscript conduct a genetic interaction degree analysis using S288C data so the ES and NES split should mirror what they are seeing for monomorphic genes and it doesn't. This needs to be explained.

Contrasting the deletion of a nonessential gene where the gene activity is completely abolished, temperature-sensitive (TS) mutations of an essential gene may only lower the gene activity partially. The potential remnant activity could lead to an underestimation of genetic interactions of the essential gene. Consequently, the true number of genetic interactions of monomorphic essential genes and that of polymorphic essential genes are likely greater than those observed in Fig. S6. It is therefore inappropriate to compare the number of genetic interactions between polymorphic essential and nonessential genes, or between monomorphic essential and nonessential genes. This is now explained in Fig. S6 legend. The reviewer's suggestion of a potential annotation bias is less of a concern because the genetic interactions used here were systematically mapped, not annotated from individual studies.

Also, authors' statement "Note that the NPI's of essential and nonessential genes are comparable because measuring protein interactions does not involve deleting genes." on page 11 par 2 is nonsensical and needs to be revised. Is it true only for polymorphic ES vs NES genes or between polymorphic and monomorphic genes or between monomorphic ES vs NES genes. Generally essential genes have more protein interactions than nonessential genes since they are hubs on the genetic and protein interaction networks. If the PPI network does mirror the GI network, this needs to be discussed formally and the associated analysis should be provided.

The word "comparable" in the quoted sentence means "can be fairly compared" rather than "similar". We have changed "comparable" to "can be fairly compared" to avoid the confusion (line 317). Additionally, here essential and nonessential genes include both monomorphic and polymorphic genes, because measuring protein interaction does not depend on whether a gene is polymorphic or monomorphic in essentiality. We agree that essential genes tend to have more PPIs than nonessential genes, as stated in our manuscript (line 293-296) and shown in Fig. 3a. Whether the PPI network mirrors the GI network has been previously studied (PMID: 15877074) and is beyond the scope of the present study.

Reviewer #3:

The authors have satisfactorily addressed my previous concerns and I am delighted to recommend the publication of the study in Nature communications.

No response is needed.